# Success History-Based Position Adaptation in Fuzzy-Controlled Ensemble of Biology-Inspired Algorithms †

**Shakhnaz Akhmedova \*** , **Vladimir Stanovov** , **Danil Erokhin and Olga Semenkina**

Department of Higher Mathematics, Reshetnev Siberian State University of Science and Technology, 660037 Krasnoyarsk, Russia; vladimirstanovov@yandex.ru (V.S.); erohhaa@mail.ru (D.E.); semenkina.olga@mail.ru (O.S.)

\*   Correspondence: shahnaz@inbox.ru

†   This paper is an extended version of our paper published in The proceedings of the 8th International Workshop on Mathematical Models and their Applications (IWMMA 2019) (Krasnoyarsk, Russian Federation, 18–21 November 2019).

**Abstract:** In this study, a new modification of the meta-heuristic approach called Co-Operation of Biology-Related Algorithms (COBRA) is proposed. Originally the COBRA approach was based on a fuzzy logic controller and used for solving real-parameter optimization problems. The basic idea consists of a cooperative work of six well-known biology-inspired algorithms, referred to as components. However, it was established that the search efficiency of COBRA depends on its ability to keep the exploitation and exploration balance when solving optimization problems. The new modification of the COBRA approach is based on other method for generating potential solutions. This method keeps a historical memory of successful positions found by individuals to lead them in different directions and therefore to improve the exploitation and exploration capabilities. The proposed technique was applied to the COBRA components and to its basic steps. The newly proposed meta-heuristic as well as other modifications of the COBRA approach and components were evaluated on three sets of various benchmark problems. The experimental results obtained by all algorithms with the same computational effort are presented and compared. It was concluded that the proposed modification outperformed other algorithms used in comparison. Therefore, its usefulness and workability were demonstrated.

**Keywords:** optimization; co-operation; biology-inspired algorithms; external archive; probabilistic distribution

## 1. Introduction

Many real-world problems can be formulated as optimization problems, which are characterized by different properties such as, for example, many local optima, non-separability, asymmetricity, etc. These problems arise from various scientific fields, such as engineering and related areas. For solving such kinds of problems, researchers have presented different methods over recent years, and heuristic optimization methods and their modifications are among them [1,2]. Random search-based and nature-inspired algorithms are faster and more efficient than traditional methods (Newton's method, bisection method, Hooke-Jeeves method, etc.) while solving high-dimensional complex multi-modal optimization problems, for example [3]. However, they also have difficulties in keeping the balance between exploration (the procedure of finding completely new areas of the search space) and exploitation (the procedure of finding the regions of a search space close to previously visited points) when solving these problems [4–6].

Biology-inspired (population-based) algorithms such as Particle Swarm Optimization (PSO) [7], Ant Colony Optimization (ACO) [8], the Artificial Bee Colony (ABC) [9], the Whale Optimization Algorithm (WOA) [10], the Grey Wolf Optimizer (GWO) [11], the Artificial Algae Algorithm (AAA) [12], Moth-Flame Optimization (MFO) [13] and others are among the most popular and frequently used heuristic optimization methods. These algorithms imitate the behavior of a group of animals (individual or social) or some of their features. Thus, the process of optimization consists of generating a set of random solutions (called individuals in other words) and leading them to the optimal solution for a given problem.

Biology-inspired algorithms have found a variety of applications for real-world problems from different areas due to their high efficiency, for example [14–16]. These algorithms gained popularity among research due to the fact that they can be used for solving various optimization problems regardless of the objective function's features. Nevertheless, according to the No Free Lunch (NFL) theorem there is no universal method for solving all optimization problems [17]. Therefore, researchers propose new optimization algorithms to increase the efficiency of the currently existing algorithms for solving a wider range of optimization problems.

One way to modify currently existing techniques consists of developing collective meta-heuristics, which use the advantages of several techniques at the same time and, therefore, are more efficient [18–20]. In [21] the meta-heuristic approach COBRA (Co-Operation of Biology-Related Algorithms) based on parallel functioning of several populations was proposed for solving unconstrained real-valued optimization problems. Its main idea can be described as the simultaneous work of several biology-inspired algorithms with similar schemes, which compete and cooperate with each other.

The original version of the COBRA consisted of six popular biology-inspired component algorithms, namely the Particle Swarm Optimization Algorithm (PSO) [7], the Cuckoo Search Algorithm (CSA) [22], the Firefly Algorithm (FFA) [23], the Bat Algorithm (BA) [24], the Fish School Search Algorithm (FSS) [25] and, finally, the Wolf Pack Search Algorithm (WPS) [26]. However, various other heuristics can be used as component algorithms (for example, the ones mentioned above) for COBRA as well as previously conducted experiments demonstrating that even the already chosen bio-inspired algorithms can be combined differently [27].

Later, the fuzzy logic-based controllers [28] were proposed for the automated selection of biology-inspired algorithms to be included in the ensemble from a predefined set, and the number of individuals in each population [29]. The idea of using fuzzy controllers for parameter adaptation of the heuristic was previously explored by researchers, for instance in [30,31], and their usefulness was established. The fuzzy-controlled COBRA modification was named COBRA-f and its efficiency was demonstrated in [32].

The COBRA-f approach, like the original COBRA algorithm, was developed for continuous optimization [29], but despite its effectiveness compared to the mentioned biology-inspired algorithms (in other words its components), the COBRA-f meta-heuristic still needs to address the problem of exploitation and exploration [33]. As was noted before, a variety of ideas has been proposed to find the exploration-exploitation balance in the population-based biology-related algorithms, including methods of parameter adaptation [34–36], island models [37,38], population size control [39,40], and many others. One of the most valuable ideas proposed for the Differential Evolution (DE) [41] algorithm in the study [42] is to use an external archive of potentially good solutions, which has limited size and updated during the optimization process. This idea is similar to the one used in multi-objective optimizers such as SPEA or SPEA2 [43], where an external archive of non-dominated solutions is maintained.

The main idea of the archive is to save promising solutions that may have valuable data about the search space and its potentially good areas, thereby highlighting the algorithms' successful search history [43]. The idea of applying this information could be used to any biology-related optimization heuristic, for instance [44,45]. In this study, the idea of applying a success history-based archive of potentially good solutions is implemented for the COBRA-f algorithm.

This paper is an extended version of our paper published in the proceedings of the 8th International Workshop on Mathematical Models and their Applications (IWMMA 2019) (Krasnoyarsk, Russian Federation, 18–21 November 2019) [46]. Algorithm introduced in [46] was tested on two additional sets of benchmark functions. Moreover, population size changes were observed while solving various benchmark problems with 10 and 30 variables. It should be noted that in this study several modifications were discussed and the number of compared algorithms increased.

Therefore, in this paper, first original COBRA meta-heuristic and its version COBRA-f are presented, and then a description of the newly proposed method for the fuzzy-controlled COBRA is presented. The next section contains the experimental results obtained by the original COBRA algorithm, the COBRA-f with fuzzy controller and the proposed approach as well as the results obtained by the COBRA's components with and without external archive are presented and discussed. The conclusions are given in the last section.

## 2. Co-Operation of Biology-Related Algorithms (COBRA)

Five biology-inspired optimization methods, to be more specific, Particle Swarm Optimization (PSO) [7], Wolf Pack Search (WPS) [26], the Firefly Algorithm (FFA) [23], the Cuckoo Search Algorithm (CSA) [22] and the Bat Algorithm (BA) [24] were used as basis for the meta-heuristic approach called Co-Operation of Biology-Related Algorithms or COBRA [21]. These algorithms are referred to as "component algorithms" of the COBRA approach. It should be noted that the number of component algorithms can be changed (increased or decreased), and it affects the workability of the COBRA meta-heuristic, which was proved in [27].

All mentioned population-based algorithms have their advantages and disadvantages. So, the possibility of using all of them simultaneously while solving any given optimization problem (namely their advantages) was the reason for the development of a new potentially better cooperative approach. Also experimental results show that it is hard to determine which algorithm should be used for a given problem, thus using a cooperative meta-heuristic means that there is no longer the necessity to choose one of the mentioned biology-inspired algorithms [21].

The optimization process of the COBRA approach starts with generating one population for each biology-inspired component algorithm, and therefore, with generating five (or six later when the Fish School Search (FSS) [25] algorithm was added to the collective) populations. After that all populations are executed in parallel or in other words are executed simultaneously, cooperating with each other.

All listed component algorithms are population-based heuristics, and thus, for each of them the population size or number of individuals (potential solutions) should be chosen beforehand, and this number does not change during the optimization process. However, the COBRA approach is a self-tuning meta-heuristic. Thus, first the minimum and maximum numbers of individuals throughout all populations are defined, and then the initial sizes of populations. Then the population size for each component algorithm changes automatically during the optimization process.

The number of individuals in the population for each component depends on the fitness values of these individuals, namely the population size can increase or decrease during the optimization process. If the overall population fitness value was not improved during a given number of iterations, then the size of each population increased, and vice versa, if the overall population fitness value was constantly improved during a given number of iterations, then the size of each population decreased. Moreover, a population size can increase by accepting individuals removed from other populations in case if its average fitness value is better than the average fitness value of all other populations. Thus, the "winner algorithm" can be determined as an algorithm whose population has the best average fitness value at every step.

The original algorithm COBRA additionally has a migration operator, which allows "communication" between the populations in ensemble. To be more specific, "communication" was determined in the following way: populations exchange individuals in such a way that a part of the

worst individuals of each population is replaced by the best individuals of other populations. Thus, the group performance of all algorithms can be improved.

The performance of the COBRA meta-heuristic approach was evaluated on a set of various benchmark and real-world problems and the experiments showed that COBRA works successfully and is reliable on different optimization problems [21]. Moreover, the simulations showed that COBRA outperforms its component algorithms when the dimension grows or when complicated problems are solved, and therefore, it should be used instead of them [21].

## 3. Fuzzy-Controlled COBRA

As was mentioned in the previous section, the original COBRA approach has six similar biology-inspired component algorithms, which mimic the collective behavior of their corresponding animal groups, thereby allowing the global optima of real-valued functions to be found. Performance analysis showed that all of them are sufficiently effective for solving optimization problems, and their workability has been established [21,47].

However, there are various other algorithms which can be used as components for COBRA as well as previously conducted experiments demonstrating that even the biology-inspired algorithms already chosen can be combined in different ways. For example, in [27] five different combinations of the population-based heuristics for the COBRA algorithm were presented, and their efficiency was examined on test problems from the CEC 2013 competition [48]. It was established that three of them show the best results on test functions depending on the number of variables [27].

The described problem was solved by controllers based on fuzzy logic [28]. The fuzzy controller implements a more flexible parameter tuning algorithm, compared to the original approach used in COBRA [29]. The fuzzy controller operates by using special fuzzification, inference and defuzzification schemes [28], which allow generating real-valued outputs. In the mentioned study [29], component algorithms are rated with success values, which were used as the fuzzy-controller inputs, and the amount of population size changes as its outputs.

The controller based on fuzzy logic used in this study had 7 inputs, including 6 success rates of component algorithms and the success rate of the whole population, and 6 outputs, including the number of individuals to add or remove from every heuristic component algorithm. The success of every component was determined as the best achieved fitness value of the corresponding component. This choice was made in accordance with the research presented in [29]. The 7-th input variable was determined as the ratio of the number of steps, during which the best-found fitness value (found by all algorithms together) was improved, to the adaptation period, which was a parameter.

To obtain the output values, the Mamdani fuzzy inference procedure was used, and the rules had the following form:

$$R_q : \text{IF } x_1 \text{ is } A_{q1} \ldots x_n \text{ is } A_{qn} \text{ THEN } y_1 \text{ is } B_{q1} \ldots y_k \text{ is } B_{qk}, \tag{1}$$

where $R_q$ is the $q$-th fuzzy rule, $x = (x_1, \ldots, x_n)$ is the set of input values in $n$-dimensional space ($n = 7$ in this case), $y = (y_1, \ldots, y_k)$ is the set of outputs ($k = 6$), $A_{qi}$ is the fuzzy set for the $i$-th input variable, $B_{qj}$ is the fuzzy set for the $j$-th output variable. The rule base consisted of 21 fuzzy rules and was structured as follows: the rules were each three rules were organized to describe the case when one of the components achieved better fitness values than the others (as there are six components, a total of 18 rules were set); the last 3 rules used the total success rate for all components (variable 7) to determine if solutions should be added or removed from all components, thus regulating the amount of available computational resources [29]. Part of the described rules base is presented in Table 1.

**Table 1.** Part of the rule base.

| Rule | | | | | | | |
|------|------|------|------|------|------|------|------|
| 1 | IF | $x_1$ is $A_3$ | $x_2$–$x_6$ is $A_4$ | $x_7$ is $DC$ | THEN | $y_1$ is $B_3$ | $y_2$–$y_6$ is $B_1$ |
| 2 | IF | $x_1$ is $A_2$ | $x_2$–$x_6$ is $A_4$ | $x_7$ is $DC$ | THEN | $y_1$ is $B_2$ | $y_2$–$y_6$ is $B_2$ |
| 3 | IF | $x_1$ is $A_1$ | $x_2$–$x_6$ is $A_4$ | $x_7$ is $DC$ | THEN | $y_1$ is $B_1$ | $y_2$–$y_6$ is $B_3$ |
| . . . | | | | . . . | | | |
| 19 | IF | | $x_1$–$x_6$ is $DC$ | $x_7$ is $A_1$ | THEN | $y_1$ is $B_1$ | |
| 20 | IF | | $x_1$–$x_6$ is $DC$ | $x_7$ is $A_2$ | THEN | $y_1$ is $B_2$ | |
| 21 | IF | | $x_1$–$x_6$ is $DC$ | $x_7$ is $A_3$ | THEN | $y_1$ is $B_3$ | |

The input variables were set to be in $[0, 1]$, and the fixed triangular fuzzy terms were used for this case. The fourth fuzzy term $A_4$ was added, "larger than 0" (opposite to $A_1$) in addition to three classical terms $A_1$, $A_2$ and $A_3$ and the "Don't Care" ($DC$) condition. The $A_4$ and '$DC$ are needed to simplify the rules and decrease their number [29].

The output variables were also set using 3 triangular fuzzy terms. These terms were symmetrical, and their positions were determined according to 2 pairs of values, which encoded the right and left positions of the central term, and the middle position of the left and right terms, and the minimal and maximal values for the side terms. These values were especially optimized with the PSO heuristic [7] and the following parameters were found: $[-12; -2; 0; 19]$ according to [29]. The defuzzification was performed by calculating using the center of mass approach for the shape obtained after the fuzzy inference.

The "communication", or in other words the migration operator, did not have any changes. The fuzzy-controlled COBRA performance was evaluated on a set of benchmark optimization problems with 10 and 30 variables from [48]. The experimental results have shown that the COBRA-f algorithm can find best solutions for many benchmark problems. Moreover, the COBRA-f meta-heuristic algorithm was compared to its components, as well as original COBRA. Thus, the simulations and the comparison have shown that the COBRA-f algorithm is superior to the previously proposed biology-related component algorithms, especially with the growth of the dimension [29].

## 4. Proposed Approach

In this study, a new modification consisting of using the success history-based position adaptation of potential solutions (SHPA) is introduced. The main idea is to improve the search diversity of biology-inspired component algorithms of the fuzzy-controlled COBRA meta-heuristic approach and consequentially COBRA's efficiency. The key concept of the proposed technique is described below.

First, one population for each component algorithm is generated, namely the set of potential solutions called individuals and represented as real-valued vectors with length $D$ is randomly generated, where $D$ is the number of dimensions for a given optimization problem. It should be noted that on this step the population size for each component is chosen beforehand and will be changed later automatically by the fuzzy controller. Also, additionally for each population (component algorithm) an external archive for best-found positions is created. At the beginning the external archive is empty and then its size can increase to the maximum value, which is chosen by the end-user and stays the same during the work of the component algorithm.

The best position found by a given individual or in other words the local best-found position in the search space for each individual in each population is saved. Initially each individual's current coordinates are used as its local best. If later a better position is found, then it will be used as the local best and the previous one will be stored in the mentioned external archive.

The pseudo-code introduced in Algorithm 1 for a minimization problem can describe the process of updating the external archive for each component algorithm.

---

**Algorithm 1** The process of updating the external archive for component algorithms

---

1:  The objective function is $f$
2:  **for** $i$ in $1 \ldots 6$ **do**
3:      $N_i$ is the size of the $i$-th population
4:      $A_i$ is the external archive for the $i$-th population
5:      $|A_i|$ is the maximum size for the $i$-th external archive
6:      $k_i$ is the current number of individuals stored in the archive $A_i$ ($k_i \leq |A_i|$)
7:      **for** $j$ in $1 \ldots k_i$ **do**
8:          $A_{ij}$ is the $j$-th individual stored in the archive $A_i$
9:      **end for**
10:     **for** $j$ in $1 \ldots N_i$ **do**
11:         $P_{ij}$ is the $j$-th individual in the $i$-th population
12:         $local_{ij}$ is the local best for each $P_{ij}$
13:     **end for**
14: **end for**
15: **for** $i$ in $1 \ldots 6$ **do**
16:     **for** $j$ in $1 \ldots N_i$ **do**
17:         **if** $f(P_{ij}) < f(local_{ij})$ **then**
18:             **if** $(k_i + 1 \leq |A_i|)$ **then**
19:                 $A_{i(k_i+1)} = local_{ij}$
20:                 $k_i = k_i + 1$
21:             **end if**
22:             **if** $(k_i + 1 > |A_i|)$ **then**
23:                 randomly choose the integer $r$ from 1 to $|A_i|$
24:                 **if** $f(local_{ij}) < f(A_{ir})$ **then**
25:                     $A_{ir} = local_{ij}$
26:                 **end if**
27:             **end if**
28:             $local_{ij} = P_{ij}$
29:         **end if**
30:     **end for**
31: **end for**

---

As was already mentioned, all component algorithms are executed in parallel after generating of six populations (one for each of them) and creation of the external archives. Thus, when individuals change their positions in the search space according to the formulas given for the considered component algorithm they can use with some probability $p_a$ the potential solutions stored in the $i$-th external archive, where $i = 1, \ldots, 6$.

It should be noted that the value of the probability $p_a$ depends on the considered biology-inspired component algorithm. More specifically, previously conducted research showed that only three components of the COBRA approach, namely the Firefly Algorithm, the Cuckoo Search Algorithm and the Bat Algorithm, demonstrate statistically better results by using an archive for the individual's position adaptation [49]. Thus, only these three algorithms use archives during their execution.

First, let us consider the Bat Algorithm [24]. Each $i$-th individual from the population in the Bat Algorithm is represented by its coordinates $x_i = (x_{i1}, \ldots, x_{iD})$ and velocity $v_i = (v_{i1}, \ldots, v_{iD})$, where $D$ is the number of dimensions of the search space. The following formulas are used for updating velocities and locations/solutions in the BA approach:

$$v_i(t + 1) = v_i(t) + (x_i(t) - x^*) \cdot f_i, \tag{2}$$

$$x_i(t+1) = v_i(t+1) + x_i(t), \tag{3}$$

where $t$ and $(t+1)$ are the numbers indicating the current and the next iterations, $x^*$ is the current best-found solution by the whole population, and $f_i$ is the frequency of the emitted pulses for the $i$-th individual [24]. Thus, with the probability $p_a$ instead of $x^*$ the randomly chosen individual from the external archive (if it is not empty) will be used. It should be noted that the external archive is also selected randomly (it is not necessarily the external archive created for the BA population). It is done with the expectation that individuals will move in multiple directions and, therefore, will be able to find better solutions.

For the other two biology-inspired component algorithms, FFA and CSA, the external archives were used in the similar way: with a given probability $p_a$ the current point of attraction $x^*$ was changed to a stored in the archive solution (from a randomly chosen archive). To be more specific, in the CSA approach individuals were sorted according to the objective function [22]. Then part of the worst ones was removed from the population and new individuals instead of them were generated by using the external archives with a given probability $p_a$. On the other hand, in the FFA approach a firefly or individual moves towards another firefly or individual if the latest has a better objective function value [23]. Thus, while using the proposed technique for the FFA approach the firefly can be moved also towards individuals from the external archives.

There are two basic steps after the simultaneous execution of all component algorithms: the fuzzy controller makes a decision about the population sizes of components (this step is called competition) and migration, or in other words the exchanging of individuals between populations (co-operation). To be more specific, the size of each population can decrease by removing some of individuals from the population to the minimal value chosen by the end-user or increase (the overall maximum size of all populations together is also established by the end-user beforehand). While increasing the population size or in other words adding new individuals, these new individuals can be generated by using the scheme in Algorithm 2.

---

**Algorithm 2** Generating of the new individuals

---

1: $padd_i$ is the probability for using the normal distribution $N(a, \sigma)$ with mean value $a$ and standard deviation $\sigma$ by the $i$-th population
2: $|A_{ci}|$ is the current archive size of the $i$-th population
3: $algbest_i$ is the currently best-found position by the $i$-th population
4: Generate random number $rand$ from the interval $[0, 1]$
5: **if** $rand \leq padd_i$ and $|A_{ci}| > 0$ **then**
6:　　Generate random integer $r$ from $[1, |A_{ci}|]$
7:　　$a = 0.5 \cdot (A_{cir} + algbest_i)$
8:　　$\sigma = |A_{cir} \smile algbest_i|$
9:　　Generate new individual $ind_{new} = N(a, \sigma)$
10: **else**
11:　　Generate new individual $ind_{new}$ around the $algbest_i$
12: **end if**

---

As was already noted, all populations communicate with each other by exchanging individuals. However, in this modification of the fuzzy-controlled COBRA, part of the worst individuals of each population is replaced by the new individuals generated by a scheme similar to the one described above (using normal distribution), but instead of $algbest_i$ the current best-found position by all populations is used and the external archive is also randomly chosen.

Thus, the proposed success history-based position adaptation method of the potential solutions depends on the probability $p_a$ (there are three values for this probability, or more specifically, one value

for each component algorithm that uses its archive during the execution), the maximum archive size $|A_i|$ and probabilities $padd_i$ (one for each component algorithm).

## 5. Results and Discussions

### 5.1. Numerical Benchmarks

To check the efficiency of the proposed algorithm, the modified fuzzy-controlled COBRA algorithm is tested on three different sets of test problems, which are 23 classical problems [50], nine standard benchmark problems with 10 and 30 variables [50], and 16 problems taken from the CEC 2014 competition [51]. These functions have been widely used in the literature [49] or [52], for example.

These functions are known as SET-1, SET-2 and SET-3, respectively. These functions are based on a set of classical benchmark functions such as Ackley's, Rastrigin's, Katsuura's, Griewank's, Weierstrass's, Sphere's, HappyCat's, Swefel's, HGBat and Rosenbrock's functions. They span a diverse set of features such as noise in the fitness function, non-separable, multimodality, ill-conditioning and rotation, among others. The functions in the stated sets of test problems are separated into three groups: unimodal, high-dimensional and low-dimensional multi-modal benchmark functions.

### 5.2. Compared Algorithms and Parametric Setup

The performance of the suggested modification of the COBRA algorithm (which will be called COBRA-SHA hereinafter) was compared with other state-of-the-art algorithms like PSO [7], WPS [26], FFA [23], CSA [22], BA [24] and FSS [25]. These algorithms have several parameters that should be initialized before running. The optimal control parameters usually depend on problems and they are unknown without prior knowledge. Therefore, the initial values of the necessary parameters for all algorithms were taken from original papers dedicated to them and proposed by authors.

Furthermore, the proposed approach was compared with modifications of the FFA, CSA and BA algorithms, which also use the external archives, as it was established previously that their usage improves the workability of the listed heuristics [48]. Let us denote them as FFA-a, CSA-a and BA-a, respectively.

To show the advantage of the proposed modification more clearly, it was also compared with the fuzzy-controlled COBRA-f [29] and also with a similar modification of the COBRA meta-heuristic, in which unlike COBRA-SHA, each component algorithm can use only its own external archive (this modification was named COBRA-fas) [53]. Parameters of the fuzzy controllers for the COBRA-fas and COBRA-SHA approaches were found by PSO in the same way as for the COBRA-f algorithm [10], namely the following parameters were obtained: $[-3; -2; 0; 10]$, $[-3; -2; 5; 10]$ and $[-12; -2; 0; 19]$ respectively. Thus, the fuzzy sets for the outputs of the obtained controllers can be represented by Figure 1.

For all mentioned biology-inspired component algorithms, the initial population size was equal to 100 on each of 23 benchmark functions from SET-1 for comparison, while the maximum number of iterations was equal to 1000. Thus, to check the efficiency of the proposed algorithm COBRA-SHA, the maximum number of function evaluations was set to $100,000$. The same number of function evaluations was used for the fuzzy-controlled COBRA-f and modification COBRA-fas. There were also 30 program runs of all algorithms, included in the comparison, for benchmark problems from SET-1.

While solving optimization problems from SET-2, the maximum generation number was 5000 and the population size for each component algorithm as well as for the FFA-a, CSA-a and BA-a modifications was equal to 100. Therefore, the maximum number of function evaluations for the COBRA-f, COBRA-fas and COBRA-SHA algorithms was set to $500,000$. It should be noted that the number of programs runs of all algorithms for benchmark problems from SET-2 was the same as for the problems from SET-1.

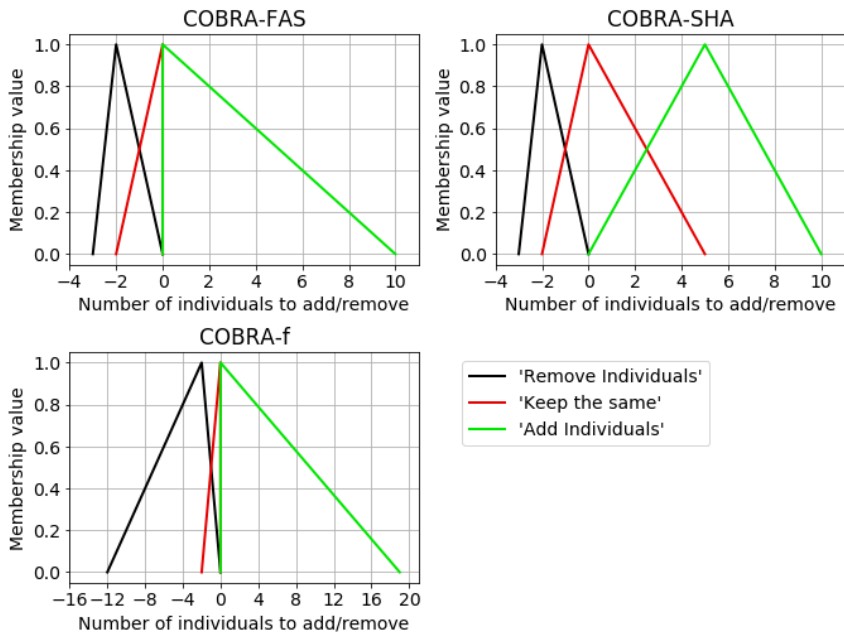

**Figure 1.** Fuzzy terms for all 6 outputs.

Finally, 16 test functions taken from the CEC 2014 Special Session on Real-Parameter Optimization [51] were solved 51 times by all mentioned heuristics. All these functions are minimization problems; they are all also shifted and scaled. The same search ranges were defined for all of them: $[-100, 100]^D$, where $D = 30$ is the number of dimensions. For all algorithms included in the comparison, the maximum number of function evaluations was equal to $300,000$. The population size for component algorithms and their modifications was set to 100.

During the experiments, the maximum archive size $A_s$ for each component of the COBRA-fas and COBRA-SHA meta-heuristics as well as for the FFA-a, CSA-a and BA-a algorithms was equal to 50. In addition, previously conducted experiments showed that the probability of using the external archive should have the following values for FFA-a, CSA-a and BA-a: 0.75, 0.6 and 0.15 respectively [49]. The same probabilities were used for the respective components of the COBRA-fas and COBRA-SHA approaches. For the rest of their component algorithms, the probability of using the external archive was set to 0 (the archive was not used specifically during the execution of a given component algorithm but was updated if conditions applied). Finally, the probability $padd_i$ for the $i$-th $(i = 1,\ldots,6)$ component algorithm of the COBRA-SHA meta-heuristic was set to 0.25.

For the collective meta-heuristic COBRA-f and its modifications mentioned in this study, while solving problems from SET-1, SET-2 and SET-3 the minimum population size for each component was set to 0, but if the total sum of population sizes was equal to 0 then all population sizes increased to 10. Additionally, the maximum total sum of population sizes was set to 300.

*5.3. Numerical Analysis on Benchmark Functions*

5.3.1. Numerical Results for SET-1

Each of the 23 problems was solved by all the stated algorithms, and experimental results such as mean value (*mean*), standard deviation (*SD*), median value (*med*) and worst (*worst*) of the best-so-far solution in the last iteration are reported. The obtained results are presented in Table 2. The outcomes, namely the mean and standard deviation values, are averaged over the number of program runs, which was equal to 30, and the best results are shown in bold type in Table 2.

**Table 2.** Minimization results of 23 benchmark functions from SET-1 for compared algorithms.

| $f$ | | PSO | WPS | FSS | CSA | FFA | BA | FFA-a | CSA-a | BA-a | COBRA-f | COBRA-fas | COBRA-SHA |
|---|---|---|---|---|---|---|---|---|---|---|---|---|---|
| 1 | mean | 1.59E−06 | 5.79E−06 | 0.000757 | 5.18E−06 | 0.00313 | 0.000388 | 0.001314 | 5.71E−07 | 7.78E−05 | 3.00E−10 | 9.75E−33 | **8.34E−104** |
| | sd | 2.43E−06 | 4.19E−06 | 7.54E−05 | 6.99E−06 | 3.73E−05 | 0.000332 | 0.000367 | 1.85E−07 | 6.51E−05 | 9.00E−10 | 5.16E−32 | **4.49E−103** |
| | med | 1.36E−07 | 5.19E−06 | 0.000719 | 1.49E−06 | 0.003123 | 0.000473 | 0.001172 | 5.39E−07 | 6.60E−05 | 7.08E−23 | 3.50E−96 | **0** |
| | worst | 5.62E−06 | 1.92E−05 | 0.000907 | 1.76E−05 | 0.003331 | 0.000851 | 0.002526 | 1.08E−06 | 0.000221 | 3.00E−09 | 2.88E−31 | **2.50E−102** |
| 2 | mean | 0.186667 | 0.227403 | 0.028172 | 0.001134 | 0.403881 | 0.035159 | 0.281052 | 0.003197 | 0.031517 | 5.06E−05 | 3.37E−32 | **9.86E−38** |
| | sd | 0.076303 | 0.012788 | 0.012715 | 0.000682 | 1.03462 | 0.022733 | 0.461951 | 0.000603 | 0.024053 | 0.000111 | 1.81E−31 | **5.31E−37** |
| | med | 0.2 | 0.225912 | 0.026171 | 0.001028 | 0.050673 | 0.031 | 0102245 | 0.002904 | 0.029492 | 5.30E−17 | 2.10E−66 | **2.09E−130** |
| | worst | 0.3 | 0.245405 | 0.054523 | 0.002339 | 5.53259 | 0.083424 | 2.30718 | 0.004141 | 0.073688 | 0.0003 | 1.01E−30 | **2.96E−36** |
| 3 | mean | 0.002586 | 0.238068 | 0.028531 | 0.079277 | 0.095334 | 0.016487 | 0.023881 | 0.003036 | 0.004284 | 3.15E−08 | 2.00E−24 | **5.79E−45** |
| | sd | 0.004729 | 0.051457 | 0.004659 | 0.095424 | 0.145768 | 0.020547 | 0.043687 | 0.001282 | 0.003749 | 1.70E−07 | 1.07E−23 | **3.12E−44** |
| | med | 0.001377 | 0.224195 | 0.029681 | 0.023966 | 0.023425 | 0.006678 | 0.010645 | 0.002862 | 0.003024 | 7.74E−31 | **6.79E−116** | 8.30E−87 |
| | worst | 0.015563 | 0.317626 | 0.033384 | 0.223789 | 0.607488 | 0.073099 | 0.238069 | 0.008804 | 0.014703 | 9.45E−07 | 5.98E−23 | **1.74E−43** |
| 4 | mean | 0.787991 | 0.225498 | 0.033573 | 0.040128 | 0.232527 | 0.002159 | 0.088004 | 0.003279 | 0.002697 | **1.77E−18** | 2.94E−14 | 2.16E−16 |
| | sd | 0.072534 | 0.037601 | 0.010726 | 0.020981 | 0.097467 | 0.001356 | 0.127233 | 0.000519 | 0.001475 | **1.73E−18** | 1.32E−13 | 1.16E−15 |
| | med | 0.763544 | 0.197098 | 0.034117 | 0.052292 | 0.200287 | 0.001925 | 0.049211 | 0.002901 | 0.002951 | 1.40E−18 | **7.62E−39** | 3.76E−25 |
| | worst | 0.904316 | 0.280872 | 0.049304 | 0.062139 | 0.667288 | 0.004693 | 0.64908 | 0.004011 | 0.004691 | **6.37E−18** | 7.32E−13 | 6.47E−15 |
| 5 | mean | 24.3935 | 26.768 | 30.9688 | 0.044777 | 32.9873 | 0.562678 | 29.8896 | **0.001852** | 0.54081 | 0.632481 | 0.709068 | 0.447074 |
| | sd | 1.17226 | 0.260772 | 0.397344 | 0.011904 | 6.95187 | 0.38868 | 2.73287 | **0.000402** | 0.334356 | 1.59647 | 0.651327 | 1.21442 |
| | med | 25.3141 | 26.7417 | 31.1182 | 0.049437 | 29.731 | 0.206465 | 28.6806 | 0.001617 | 0.547022 | 0.076141 | 0.396998 | **6.09E−06** |
| | worst | 25.4328 | 27.1092 | 31.1302 | 0.051008 | 58.5146 | 0.981097 | 39.7638 | **0.002882** | 0.975569 | 7.06745 | 2.79167 | 5.06674 |
| 6 | mean | 4.37E−07 | **0** | **0** | 0.00084 | 0.069408 | **0** | 0.035759 | 0.000856 | **0** | **0** | **0** | **0** |
| | sd | 3.80E−07 | **0** | **0** | 0.000306 | 0.001029 | **0** | 0.011896 | 0.000246 | **0** | **0** | **0** | **0** |
| | med | 5.39E−07 | **0** | **0** | 0.000732 | 0.069306 | **0** | 0.032395 | 0.001038 | **0** | **0** | **0** | **0** |
| | worst | 1.21E−06 | **0** | **0** | 0.001217 | 0.071567 | **0** | 0.062217 | 0.001038 | **0** | **0** | **0** | **0** |
| 7 | mean | 0.022798 | 0.011072 | 0.034065 | 0.001386 | 0.119858 | 0.000363 | 0.023482 | 0.0003 | 0.000283 | 0.000971 | **0.000164** | 0.000183 |
| | sd | 0.009644 | 0.006379 | 0.006115 | 0.000726 | 0.024429 | 0.000916 | 0.009803 | **7.16E−05** | 0.000455 | 0.002437 | 0.000184 | 0.000123 |
| | med | 0.018051 | 0.013906 | 0.0354315 | 0.001851 | 0.11158 | 0.000172 | 0.02019 | 0.000353 | 9.16E−05 | 0.000148 | **0.000105** | 0.000172 |
| | worst | 0.041091 | 0.02156 | 0.045658 | 0.002248 | 0.173204 | 0.005127 | 0.060796 | **0.000465** | 0.001617 | 0.009513 | 0.000944 | 0.000562 |
| 8 | mean | −3365.88 | −3715.84 | −1953.42 | −3833.45 | −2004.41 | −4113.93 | −2233.67 | **−4189.83** | −4095.29 | −4080.25 | −4129.2 | −4187.24 |
| | sd | 348.708 | 0.365664 | 349.539 | 102.134 | 34.4298 | 241.696 | 218.254 | **0** | 340.29 | 297.391 | 325.696 | 13.9255 |
| | med | −3597.64 | −3715.98 | −1924.47 | −3833 | −2004.41 | **−4189.83** | −2291.01 | **−4189.83** | **−4189.83** | **−4189.83** | **−4189.83** | **−4189.83** |
| | worst | −2999.38 | −3714.49 | −1596.36 | −3594.6 | −1969.98 | −3107.78 | −1300.58 | **−4189.83** | −2709.11 | −2973.94 | −2375.27 | −4112.25 |
| 9 | mean | 25.1497 | 0.443485 | 25.9596 | 0.003443 | 31.1039 | 0.014076 | 21.6072 | 1.42E−05 | 0.012228 | **0** | **0** | **0** |
| | sd | 14.569 | 0.150292 | 9.92443 | 0.000182 | 7.9054 | 0.014305 | 2.1656 | 2.47E−06 | 0.013029 | **0** | **0** | **0** |
| | med | 23.3826 | 0.476669 | 23.3999 | 0.003429 | 26.381 | 0.008969 | 21.7469 | 1.46E−05 | 0.006788 | **0** | **0** | **0** |
| | worst | 44.9872 | 0.612069 | 40.5318 | 0.003724 | 44.8781 | 0.046842 | 26.2129 | 1.46E−05 | 0.039491 | **0** | **0** | **0** |

**Table 2.** *Cont.*

| $f$ | | PSO | WPS | FSS | CSA | FFA | BA | FFA-a | CSA-a | BA-a | COBRA-f | COBRA-fas | COBRA-SHA |
|---|---|---|---|---|---|---|---|---|---|---|---|---|---|
| 10 | mean | 2.33524 | 0.286503 | 0.022724 | 0.007219 | 2.19393 | **−4.44E−16** | 1.87908 | 0.001914 | **−4.44E−16** | **−4.44E−16** | **−4.44E−16** | **−4.44E−16** |
| | sd | 5.22105 | 0.107099 | 0.006266 | 0.033985 | 0.56606 | **0** | 0.675231 | 0.000136 | **0** | **0** | **0** | **0** |
| | med | 0.000238 | 0.233252 | 0.022854 | 0.000711 | 2.0135 | **−4.44E−16** | 1.64645 | 0.001889 | **−4.44E−16** | **−4.44E−16** | **−4.44E−16** | **−4.44E−16** |
| | worst | 14.0099 | 0.498479 | 0.032924 | 0.190152 | 4.68214 | **−4.44E−16** | 4.78962 | 0.002645 | **−4.44E−16** | **−4.44E−16** | **−4.44E−16** | **−4.44E−16** |
| 11 | mean | 0.022367 | 0.667855 | 0.02914 | 0.005382 | 1.3761 | 6.23E−06 | 0.330717 | 0.004404 | 7.38E−06 | 6.68E−12 | **0** | **0** |
| | sd | 0.012069 | 0.06976 | 0.007509 | 0.00053 | 0.1058 | 4.95E−06 | 0.249647 | 0.000421 | 6.54E−06 | 3.60E−11 | **0** | **0** |
| | med | 0.017244 | 0.664388 | 0.03115 | 0.005528 | 1.33741 | 5.67E−06 | 0.246232 | 0.004419 | 5.33E−06 | **0** | **0** | **0** |
| | worst | 0.041632 | 0.762223 | 0.040675 | 0.00609 | 1.68613 | 1.65E−05 | 1.44168 | 0.005244 | 2.27E−05 | 2.00E−10 | **0** | **0** |
| 12 | mean | 0.832804 | 0.002054 | 0.030286 | 0.007105 | 1.23821 | 0.285361 | 1.11234 | **1.62E−05** | 0.165609 | 0.000471 | 0.00015 | 4.74E−05 |
| | sd | 2.4112 | 0.000881 | 0.010161 | 0.025964 | 0.474751 | 0.303834 | 0.538147 | **8.53E−06** | 0.136602 | 0.001761 | 0.000607 | 0.000253 |
| | med | 0.032622 | 0.001947 | 0.027299 | 2.22E−05 | 1.09173 | 0.15708 | 0.914424 | 1.48E−05 | 0.144359 | 2.45E−05 | 3.37E−05 | **8.97E−16** |
| | worst | 8.0664 | 0.003608 | 0.054965 | 0.104246 | 3.43408 | 1.1781 | 3.49961 | **3.61E−05** | 0.543077 | 0.009578 | 0.003419 | 0.001412 |
| 13 | mean | 0.069719 | 0.038975 | 0.030959 | 2.46E−05 | 0.836476 | 0.292157 | 0.477642 | 1.47E−05 | 0.399484 | 0.002164 | 0.001114 | **1.15E−09** |
| | sd | 0.134767 | 0.017712 | 0.013819 | 8.74E−05 | 0.120599 | 0.204197 | 0.239367 | 7.57E−06 | 0.206086 | 0.00597 | 0.004535 | **3.02E−09** |
| | med | 0.000175 | 0.048161 | 0.033939 | 3.92E−06 | 0.801098 | 0.398078 | 0.403263 | 1.03E−05 | 0.144359 | 0.000442 | 3.14E−16 | **2.08E−18** |
| | worst | 0.521717 | 0.065703 | 0.052993 | 0.000492 | 1.20327 | 0.799661 | 1.40009 | 2.66E−05 | 0.798164 | 0.030962 | 0.02375 | **1.04E−08** |
| 14 | mean | **0.998** | 0.998 | 1.5642 | 0.998 | 1.9947 | 0.998 | 1.01566 | 0.998 | 0.998 | **0.998** | **0.998** | **0.998** |
| | sd | **0** | 7.08E−12 | 0.892633 | 1.57E−16 | 0.979812 | 1.75E−16 | 0.06081 | 9.90E−16 | 4.74E−16 | **0** | **0** | **0** |
| | med | **0.998** | 0.998 | 1.0605 | 0.998 | 1.992 | 0.998 | 0.998 | 0.998 | 0.998 | **0.998** | **0.998** | **0.998** |
| | worst | **0.998** | 0.998 | 3.9686 | 0.998 | 6.9034 | 0.998 | 1.24136 | 0.998 | 0.998 | **0.998** | **0.998** | **0.998** |
| 15 | mean | 0.000641 | 0.005924 | 0.000524 | 0.000366 | 0.001927 | 0.003983 | 0.000864 | 0.000324 | 0.003981 | 0.000566 | 0.00031 | **0.000307** |
| | sd | 0.000216 | 0.008707 | **2.17E−19** | 8.05E−05 | 0.000119 | 0.002392 | 0.000977 | 6.55E−05 | 0.002794 | 0.000269 | 1.13E−05 | 1.12E−07 |
| | med | 0.000783 | 0.000673 | 0.000524 | 0.000365 | 0.001975 | 0.002992 | 0.000558 | 0.000308 | 0.002615 | 0.000389 | 0.000307 | **0.000307** |
| | worst | 0.000783 | 0.020363 | 0.000524 | 0.000783 | 0.002242 | 0.008416 | 0.005758 | 0.000653 | 0.008663 | 0.000941 | 0.000369 | **0.000308** |
| 16 | mean | **−1.0316** | −1.0316 | −1.0316 | −1.0316 | −1.0281 | −1.0316 | −1.0316 | −1.0316 | −1.0316 | **−1.0316** | **−1.0316** | −1.0316 |
| | sd | **0** | 1.97E−07 | 6.66E−16 | 2.69E−05 | 0.00149 | 5.51E−16 | 3.97E−06 | 5.20E−08 | 2.88E−07 | **0** | **0** | **0** |
| | med | **−1.0316** | −1.0316 | −1.0316 | −1.0316 | −1.0268 | −1.0316 | −1.0316 | −1.0316 | −1.0316 | **−1.0316** | **−1.0316** | −1.0316 |
| | worst | **−1.0316** | −1.0316 | −1.0316 | −1.0315 | −1.0268 | −1.0316 | −1.0316 | −1.0316 | −1.0316 | **−1.0316** | **−1.0316** | −1.0316 |
| 17 | mean | **0.39789** | 0.39789 | **0.39789** | 0.398604 | 0.39809 | 0.39789 | 0.39789 | 0.39789 | 0.39789 | **0.39789** | 0.39789 | 0.39789 |
| | sd | **0** | 1.34E−08 | **0** | 0.001596 | 2.64E−08 | 2.01E−15 | 2.12E−05 | 5.42E−15 | 3.19E−16 | **0** | **0** | **0** |
| | med | **0.39789** | 0.39789 | **0.39789** | 0.397918 | 0.39809 | 0.39789 | 0.39789 | 0.39789 | 0.39789 | **0.39789** | 0.39789 | 0.39789 |
| | worst | **0.39789** | 0.39789 | **0.39789** | 0.403215 | 0.39809 | 0.39789 | 0.398006 | 0.39801 | 0.39789 | **0.39789** | 0.39789 | 0.39789 |
| 18 | mean | 3.00E+00 | 3.00001 | 3.0145 | 3.081 | 3.02798 | 3.00E+00 | 3.00011 | 3.01732 | 3.00E+00 | **3.00E+00** | **3.00E+00** | **3.00E+00** |
| | sd | 2.22E−15 | 1.41E−06 | 8.88E−16 | 0.032441 | 0.081617 | 2.86E−14 | 0.000416 | 0.003692 | 1.42E−14 | **0** | **0** | **0** |
| | med | 3.00E+00 | 3.00E+00 | 3.0145 | 3.0579 | 3.0128 | 3.00E+00 | 3.00E+00 | 3.0184 | 3.00E+00 | **3.00E+00** | **3.00E+00** | **3.00E+00** |
| | worst | 3.00E+00 | 3.00001 | 3.0145 | 3.1464 | 3.4675 | 3.00E+00 | 3.00217 | 3.0184 | 3.00E+00 | **3.00E+00** | **3.00E+00** | **3.00E+00** |

**Table 2.** *Cont.*

| $f$ | | PSO | WPS | FSS | CSA | FFA | BA | FFA-a | CSA-a | BA-a | COBRA-f | COBRA-fas | COBRA-SHA |
|---|---|---|---|---|---|---|---|---|---|---|---|---|---|
| 19 | mean | −3.8628 | −3.8628 | −3.8617 | −3.8627 | −3.8312 | −3.2190 | −3.8627 | −3.8472 | −3.8628 | **−3.8628** | **−3.8628** | −3.8628 |
| | sd | 3.11E−15 | 2.76E−07 | 0.000558 | 8.61E−05 | 0.022515 | 0.638175 | 0.00043 | 0.016026 | 1.46E−06 | **0** | **0** | 3.44E−06 |
| | med | −3.8628 | −3.8628 | −3.8619 | −3.8627 | −3.8328 | −3.7754 | −3.8628 | −3.8614 | −3.8628 | **−3.8628** | **−3.8628** | −3.8628 |
| | worst | −3.8628 | −3.8628 | −3.8609 | −3.8623 | −3.7831 | −2.2391 | −3.8604 | −3.8257 | −3.8628 | **−3.8628** | **−3.8628** | −3.8628 |
| 20 | mean | −3.2429 | **−3.3223** | −3.3137 | −3.2012 | −3.2179 | −3.3192 | −3.2908 | −3.2523 | **−3.3223** | −3.3221 | **−3.3223** | **−3.3223** |
| | sd | 0.056194 | **5.58E−05** | 0.002282 | 0.001506 | 0.371432 | 0.016419 | 0.117259 | 0.078632 | 8.15E−05 | 0.001024 | 0.000126 | 0.000142 |
| | med | −3.2032 | −3.3222 | −3.3121 | −3.2017 | −3.3223 | −3.3222 | −3.3224 | −3.3121 | −3.3223 | −3.3223 | **−3.3224** | **−3.3224** |
| | worst | −3.2032 | **−3.3222** | −3.3121 | −3.1964 | −1.3854 | −3.2307 | −2.684 | −3.1471 | −3.322 | −3.3166 | −3.3217 | −3.3216 |
| 21 | mean | −4.4259 | −10.1532 | −10.147 | −10.1531 | −9.67012 | −10.153 | −10.1231 | −10.1532 | −9.74075 | −10.153 | −10.153 | **−10.1532** |
| | sd | 3.1596 | 1.64E−05 | 8.88E−15 | 4.68E−05 | 1.21432 | 0.000913 | 0.138499 | 2.23E−05 | 1.55633 | 0 | 0.001217 | **0** |
| | med | −2.6829 | −10.1532 | −10.147 | −10.1531 | −10.1387 | −10.1532 | −10.1532 | −10.1532 | −10.1532 | −10.153 | −10.1532 | **−10.1532** |
| | worst | −2.6829 | −10.1532 | −10.147 | −10.153 | −4.07347 | −10.1481 | −9.38354 | −10.1531 | −2.56105 | −10.153 | −10.1464 | **−10.1532** |
| 22 | mean | −5.2731 | −10.4029 | −10.3966 | −10.4027 | −9.89302 | −9.46839 | −10.3104 | −10.4029 | −9.85423 | −10.4028 | −10.4029 | **−10.4029** |
| | sd | 0.952733 | 6.63E−06 | 0.002702 | 0.000178 | 0.100572 | 2.11049 | 0.406034 | 2.20E−05 | 1.59768 | 0.000272 | 5.62E−05 | **0** |
| | med | −5.0877 | −10.4029 | −10.3964 | −10.4028 | −9.9117 | −10.4029 | −10.4029 | −10.4029 | −10.4029 | −10.4029 | −10.4029 | **−10.4029** |
| | worst | −5.0877 | −10.4029 | −10.3937 | −10.4021 | −9.35142 | −3.55911 | −8.14783 | −10.4028 | −5.02718 | −10.4014 | −10.4026 | **−10.4029** |
| 23 | mean | −4.0308 | −10.5364 | −10.531 | −10.5362 | −9.37719 | −10.2025 | −10.4969 | −10.5364 | −10.5309 | −10.5227 | **−10.5364** | **−10.5364** |
| | sd | 2.9128 | 1.07E−05 | 0.002013 | 0.000154 | 1.6799 | 1.20832 | 0.16659 | 5.05E−05 | 0.029262 | 0.073438 | **0** | **0** |
| | med | −2.8066 | −10.5364 | −10.53 | −10.5362 | −10.0336 | −10.5364 | −10.5364 | −10.5364 | −10.5363 | −10.5364 | **−10.5364** | **−10.5364** |
| | worst | −2.4217 | −10.5364 | −10.53 | −10.5359 | −4.41148 | −4.61133 | −9.61849 | −10.5362 | −10.3733 | −10.1272 | **−10.5364** | **−10.5364** |

From Table 2 it can be observed that the proposed approach COBRA-SHA outperformed other compared state-of-art approaches and their modifications as well as COBRA-f and the similar modification COBRA-fas on the first two unimodal functions ($f_1$ and $f_2$) in terms of the mean, standard deviation, median, and worst value of the results. Regarding function $f_3$, COBRA-SHA was outperformed only by the modification COBRA-fas in terms of the median value, while it was the best among the compared algorithms according to the other statistical results.

The fuzzy-controlled COBRA outperformed the other algorithms on the function $f_4$. Regarding the fifth unimodal function, while the CSA modification with the external archive demonstrated the best results in terms of the mean, standard deviation and the worst values, the median value obtained by the proposed approach COBRA was better. Several algorithms, including COBRA-f, COBRA-fas and COBRA-SHA, were able to find the optimum value for the function $f_6$ during each program run. Finally, regarding function $f_7$, COBRA-fas and CSA-a outperformed the other algorithms.

For multi-modal functions $f_8$–$f_{13}$ with many local minima, the final results are more important because this function can reflect the algorithm's ability to escape from poor local optima and obtain the near global optimum. For functions $f_9$, $f_{10}$ and $f_{11}$, COBRA-SHA was successful in finding the global minimum as well as the fuzzy-controlled COBRA and the similar modification COBRA-fas. For function $f_8$, CSA with the external archive (CSA-a) outperformed the other algorithms included in the comparison. Regarding $f_{12}$, the proposed approach COBRA-SHA was the best in terms of the median value, while CSA-a outperformed all the compared algorithms according to the other statistical results. Moreover, for functions $f_{13}$ the proposed modification COBRA-SHA produced better results compared to the others.

For $f_{14}$–$f_{23}$ with only a few local minima, the dimension of the function is also small. For functions $f_{14}$, $f_{16}$, $f_{17}$, $f_{18}$, $f_{21}$, $f_{22}$ and $f_{23}$, COBRA-SHA was successful in finding the global minimum. Regarding $f_{14}$ and $f_{16}$, PSO, COBRA-f, COBRA-fas and COBRA-SHA produced the same results. For function $f_{17}$, PSO, FSS, COBRA-f, COBRA-fas and COBRA-SHA also gave the same values. Regarding $f_{18}$ COBRA-f, COBRA-fas and COBRA-SHA produced the same mean, standard deviation, median and worst values. Finally, for function $f_{23}$ the two similar modifications proposed in this study, namely COBRA-fas and COBRA-SHA, demonstrated the same results.

From Table 2, it can be observed that the COBRA-SHA approach performs better than the other algorithms on the multi-modal low-dimensional benchmarks. For example, regarding function $f_{15}$, the COBRA-SHA approach outperformed other algorithms included in the comparison in terms of the mean, median and worst values. However, for function $f_{19}$ COBRA-f and COBRA-fas were able to find the optimum value during each program run and they outperformed COBRA-SHA. Finally, regarding function $f_{20}$, the best mean and median values were found by the proposed modifications COBRA-fas and COBRA-SHA.

Additionally, in Table 3 the results of the comparison between COBRA-SHA and the other mentioned algorithms according to the Mann-Whitney statistical test with significance level $p = 0.01$ are presented. The following notations are used in Table 3: "+" means that COBRA-SHA was better compared to a given algorithm, similarly "−" means that the proposed algorithm was statistically worse, and "=" means that there was no significant difference between their results.

**Table 3.** Results of the Mann-Whitney statistical test with $p = 0.01$ for SET-1, comparison of COBRA-SHA with other approaches.

|       | PSO | WPS | FSS | CSA | FFA | BA | FFA-a | CSA-a | BA-a | COBRA-f | COBRA-fas |
|-------|-----|-----|-----|-----|-----|----|-------|-------|------|---------|-----------|
| +     | 18  | 15  | 20  | 22  | 23  | 14 | 18    | 16    | 12   | 10      | 4         |
| =     | 5   | 8   | 3   | 0   | 0   | 9  | 5     | 7     | 11   | 13      | 18        |
| −     | 0   | 0   | 0   | 1   | 0   | 0  | 0     | 0     | 0    | 0       | 1         |
| Total | 18  | 15  | 20  | 22  | 23  | 14 | 18    | 16    | 12   | 10      | 3         |

The results of the Mann-Whitney statistical test are presented in Figure 2. The values on the graph represent the total score, i.e., number of improvements, deteriorations and non-significant differences between COBRA-SHA and other approaches.

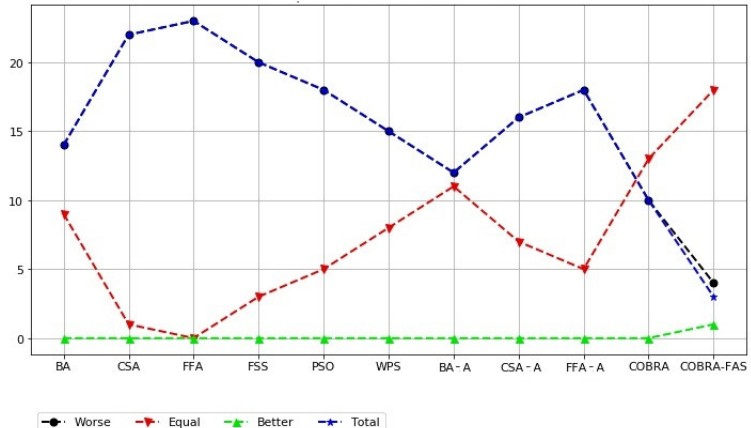

**Figure 2.** Results of the Mann-Whitney statistical test with $p = 0.01$, comparison of COBRA-SHA with other approaches (SET-1).

In addition, all the mentioned algorithms were compared with the proposed modification COBRA-SHA according to the Friedman statistical test. The obtained results are demonstrated in Figure 3. The following notations were used in Figure 3: COBRA-f was denoted as "COBRA", COBRA-fas was denoted as "C-FAS" and for COBRA-SHA the notation "C ARC" was used. The Friedman ranking was performed for every test function separately and used the results of all runs for ranking.

Thus, it was established that the results obtained by the proposed approach are statistically better according to the Friedman and Mann-Whitney tests than the results obtained by the stated biology-inspired algorithms (PSO, WPS, FSS, FFA, CSA and BA) and their modifications with the external archive (FFA-a, CSA-a, BA-a). Despite this, it can be seen that the results achieved by FFA-a, CSA-a and BA-a are statistically better than the ones found by their original versions. Moreover, COBRA-SHA statically outperformed the fuzzy-controlled COBRA-f. However, there is almost no difference between the results obtained by COBRA-SHA and the similar modification COBRA-fas on functions from SET-1.

### 5.3.2. Numerical Results for SET-2

To show the advantage of the proposed modification COBRA-SHA more clearly, it was compared with the same algorithms (mentioned previously) by using benchmark functions from SET-2. The functions used in SET-2 are Sphere, Rosenbrock, Quadric, Schwefel, Griewank, Weierstrass, Quartic, Rastrigin and Ackley, which are frequently used benchmark functions to test the performance of various optimization algorithms. These functions can be described as continuous, differentiable, separable, scalable and multi-modal.

The experimental results obtained for 10- and 30-dimensional functions by the listed biology-inspired algorithms and their modifications are shown in Tables 4 and 5. From these tables, it can be observed that the COBRA-SHA approach performs better than the other algorithms included in the comparison.

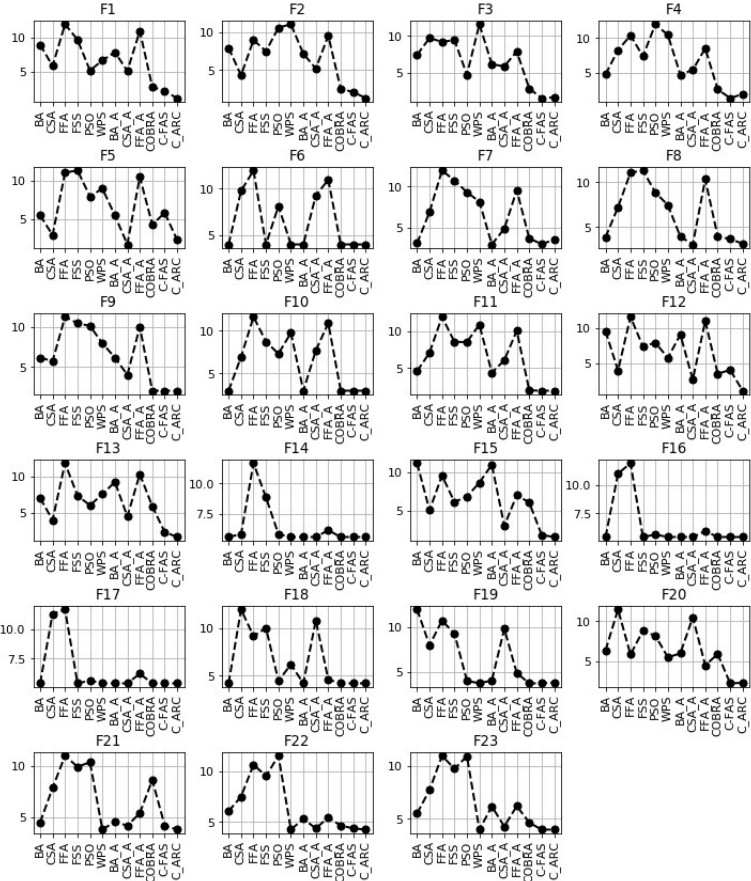

**Figure 3.** Results of the Friedman statistical test for SET-1.

For example, regarding function $f_1$, the COBRA-SHA approach outperformed the other algorithms included in the comparison in terms of the mean, best and worst values when $D = 10$. However, for the same function with $D = 30$ COBRA-f was able to find the best value during 51 program runs, while COBRA-SHA was still better than the others in terms of the mean and worst values. Similarly, for function $f_3$ the best value was found by COBRA-f, and COBRA-SHA was able to achieve better mean and worst values both with $D = 10$ and with $D = 30$.

Regarding functions $f_5$ and $f_8$ with 10 and 30 variables, COBRA-f, COBRA-fas and COBRA-SHA were able to find the optimum solutions during each program run. It should be noted that for function $f_4$ with 10 variables, the proposed modifications CSA-a, COBRA-fas and COBRA-SHA also achieved the optimum value during each program run, while the modification BA-a and the original algorithm COBRA-f found the optimum several times. On the other hand, for the same function but with 30 variables COBRA-SHA outperformed the other algorithms included in comparison. Additionally, for the last function $f_9$ both with $D = 10$ and with $D = 30$ COBRA-f, COBRA-fas, COBRA-SHA, BA and its modification BA-a demonstrated the same good results.

As for the second function $f_2$ ($D = 10$ and $D = 30$), CSA-a outperformed the other algorithms in terms of mean and worst values, but the best value was found by the COBRA-fas approach. Regarding function $f_6$ with 10 variables, the PSO algorithm demonstrated the best results, while for that benchmark problem with 30 variables COBRA-fas outperformed every algorithm included in comparison. Finally, for function $f_7$ with $D = 10$, BA and BA-a gave better results, and with $D = 30$ COBRA-SHA did the same.

Table 4. Experimental results for 10-dimensional functions from SET-2.

| $f$ | | PSO | WPS | FSS | CSA | FFA | BA | FFA-a | CSA-a | BA-a | COBRA-f | COBRA-fas | COBRA-SHA |
|---|---|---|---|---|---|---|---|---|---|---|---|---|---|
| 1 | best | 1.63E−24 | 8.63E−08 | 0.000161 | 6.56E−08 | 0.000599 | 6.67E−10 | 0.000273 | 4.90E−08 | 4.75E−13 | **0** | **0** | **0** |
| | worst | 3.51E−24 | 3.69E−06 | 0.000403 | 1.70E−07 | 0.002639 | 4.44E−06 | 0.001723 | 1.50E−07 | 3.99E−06 | 2.81E−15 | 6.47E−16 | **1.05E−223** |
| | mean | 8.60E−24 | 8.42E−07 | 0.00026 | 1.07E−07 | 0.000995 | 1.10E−06 | 0.000678 | 1.05E−07 | 9.46E−07 | 1.88E−16 | 2.16E−17 | **3.58E−225** |
| | sd | 1.06E−23 | 9.00E−07 | 9.64E−05 | 3.02E−08 | 0.00058 | 1.15E−06 | 0.000405 | 2.64E−08 | 1.11E−06 | 7.02E−16 | 1.16E−16 | **0** |
| 2 | best | 0.049514 | 0.213844 | 7.40186 | 0.001172 | 11.4098 | 0.089051 | 10.7438 | 0.000126 | 0.088981 | 0.044043 | **7.54E−24** | 1.25E−11 |
| | worst | 0.17305 | 0.330575 | 8.46311 | 0.010075 | 20.0667 | 0.089954 | 16.0295 | **0.001501** | 0.089981 | 0.189479 | 1.7902 | 0.015126 |
| | mean | 0.082267 | 0.257968 | 7.99109 | 0.005168 | 15.6137 | 0.0893 | 13.452 | **0.000679** | 0.089217 | 0.099337 | 0.063924 | 0.001354 |
| | sd | 0.041129 | 0.043318 | 0.393817 | 0.00263 | 3.023 | **0.000287** | 1.96209 | 0.000373 | 0.000273 | 0.031152 | 0.320857 | 0.003361 |
| 3 | best | 1.45E−17 | 1.70E−15 | 0.000484 | 7.54E−05 | 0.027544 | 0.001073 | 0.007773 | 3.83E−05 | 0.000231 | **0** | 6.29E−170 | **0** |
| | worst | 6.19E−16 | 3.77E−05 | 0.001354 | 0.00029 | 0.092337 | 0.009179 | 0.051593 | 0.000246 | 0.007369 | 1.39E−11 | 4.25E−13 | **5.67E−104** |
| | mean | 1.65E−16 | 2.66E−06 | 0.000959 | 0.000193 | 0.058774 | 0.005257 | 0.025486 | 0.000163 | 0.003161 | 7.51E−13 | 1.54E−14 | **1.89E−105** |
| | sd | 1.72E−16 | 9.38E−06 | 0.000304 | 5.54E−05 | 0.026681 | 0.00191 | 0.015622 | 5.10E−05 | 0.0019 | 2.89E−12 | 7.64E−14 | **1.02E−104** |
| 4 | best | 118.438 | 236.89 | 1870.76 | 236.88 | 1970.439 | 16.029 | 1773.079 | **0** | **0** | **0** | **0** | **0** |
| | worst | 950.541 | 830.59 | 2658.19 | 1191.97 | 2807.619 | 507.569 | 2882.499 | **0** | 474.63 | 1899.23 | **0** | **0** |
| | mean | 526.392 | 584.91 | 2217.53 | 586.12 | 2436.108 | 333.146 | 2387.664 | **0** | 63.226 | 65.36 | **0** | **0** |
| | sd | 227.166 | 152.965 | 254.724 | 316.432 | 292.505 | 136.605 | 313.144 | **0** | 132.54 | 340.72 | **0** | **0** |
| 5 | best | 0.017241 | 0.013663 | 0.004056 | 5.71E−05 | 0.114819 | 1.99E−09 | 0.020086 | 4.81E−05 | 2.76E−09 | **0** | **0** | **0** |
| | worst | 0.076242 | 0.074725 | 0.008699 | 0.000168 | 0.31823 | 4.46E−08 | 0.038537 | 0.000151 | 2.72E−08 | **0** | **0** | **0** |
| | mean | 0.052542 | 0.043517 | 0.006624 | 0.000109 | 0.209934 | 1.04E−08 | 0.029265 | 9.81E−05 | 8.10E−09 | **0** | **0** | **0** |
| | sd | 0.017733 | 0.013805 | 0.001311 | 2.35E−05 | 0.058816 | 1.19E−08 | 0.00484 | 3.10E−05 | 6.80E−09 | **0** | **0** | **0** |
| 6 | best | **−7.15E−09** | −9.76E−10 | 0.001233 | 0.000745 | 0.052097 | 0.071887 | 0.038534 | 0.000739 | 0.016961 | −7.06E−09 | **−7.15E−09** | −1.59E−09 |
| | worst | **−7.15E−09** | 6.46E−13 | 0.002208 | 0.001241 | 0.07805 | 0.281316 | 0.062414 | 0.001121 | 0.290965 | 1.06E−14 | −7.36E−10 | −3.69E−10 |
| | mean | **−7.15E−09** | −2.27E−10 | 0.001515 | 0.001067 | 0.063843 | 0.146193 | 0.050823 | 0.000984 | 0.13815 | −1.21E−09 | −3.28E−09 | −1.26E−09 |
| | sd | **0** | 3.1E−10 | 0.000201 | 0.000105 | 0.006625 | 0.045155 | 0.006592 | 9.61E−05 | 0.059034 | 2.28E−09 | 2.07E−09 | 4.01E−10 |
| 7 | best | 3.59E−05 | 9.19E−05 | 0.000943 | 4.12E−05 | 0.010896 | 2.98E−11 | 0.001142 | 2.35E−05 | **6.21E−12** | 9.63E−11 | 1.05E−09 | 3.60E−07 |
| | worst | 0.00044 | 0.002463 | 0.00362 | 0.000545 | 0.079071 | **0.000104** | 0.011006 | 0.000295 | 0.000122 | 0.001366 | 0.000889 | 0.000625 |
| | mean | 0.000172 | 0.000811 | 0.002205 | 0.000251 | 0.025808 | **2.53E−05** | 0.00501 | 0.000112 | 4.31E−05 | 0.000187 | 0.000106 | 3.87E−05 |
| | sd | 0.000105 | 0.000599 | 0.000645 | 0.000121 | 0.016032 | **2.81E−05** | 0.00313 | 5.91E−05 | 3.97E−05 | 0.000277 | 0.00018 | 0.000117 |
| 8 | best | **0** | 1.59E−05 | 0.079887 | 0.000398 | 1.16538 | 1.22E−05 | 0.913614 | 4.70E−06 | 6.98e−06 | **0** | **0** | **0** |
| | worst | 5.96975 | 0.000482 | 4.11265 | 0.000897 | 3.77051 | 0.004259 | 3.10652 | 8.42E−06 | 0.002881 | **0** | **0** | **0** |
| | mean | 1.49244 | 0.000154 | 1.46415 | 0.000688 | 2.67899 | 0.001166 | 1.92521 | 6.88E−06 | 0.000892 | **0** | **0** | **0** |
| | sd | 1.42456 | 0.000140 | 1.06661 | 0.000101 | 0.882195 | 0.001023 | 0.679404 | 1.05E−06 | 0.000828 | **0** | **0** | **0** |
| 9 | best | 6.33E−10 | 0.003801 | 0.023337 | 0.000409 | 0.308134 | **−4.44E−16** | 0.336958 | 0.000284 | **−4.44E−16** | **−4.44E−16** | **−4.44E−16** | **−4.44E−16** |
| | worst | 4.37E−09 | 0.009605 | 0.053186 | 0.000601 | 0.622152 | **−4.44E−16** | 0.662198 | 0.000501 | **−4.44E−16** | **−4.44E−16** | **−4.44E−16** | **−4.44E−16** |
| | mean | 2.30E−09 | 0.006637 | 0.037914 | 0.000561 | 0.551782 | **−4.44E−16** | 0.473258 | 0.000432 | **−4.44E−16** | **−4.44E−16** | **−4.44E−16** | **−4.44E−16** |
| | sd | 1.08E−09 | 0.001927 | 0.00896 | 4.21E−05 | 0.078404 | **0** | 0.093197 | 5.53E−05 | **0** | **0** | **0** | **0** |

**Table 5.** Experimental results for 30-dimensional functions from SET-2.

| $f$ | | PSO | WPS | FSS | CSA | FFA | BA | FFA-a | CSA-a | BA-a | COBRA-f | COBRA-fas | COBRA-SHA |
|---|---|---|---|---|---|---|---|---|---|---|---|---|---|
| 1 | best | 7.48E−09 | 6.36E−06 | 0.000678 | 7.94E−07 | 0.007467 | 3.26E−07 | 0.003507 | 3.28E−07 | 1.53E−07 | **0** | 3.50E−245 | **0** |
| | worst | 4.42E−07 | 8.57E−05 | 0.002339 | 1.61E−06 | 0.012457 | 7.42E−06 | 0.00799 | 1.85E−06 | 5.68E−06 | 1.39E−09 | 9.89E−17 | **2.12E−153** |
| | mean | 8.05E−08 | 2.57E−05 | 0.001251 | 1.21E−06 | 0.009875 | 3.15E−06 | 0.006719 | 9.97E−07 | 2.32E−06 | 5.22E−11 | 5.10E−18 | **7.08E−155** |
| | sd | 1.17E−07 | 1.98E−05 | 0.000545 | 2.23E−07 | 0.001831 | 2.10E−06 | 0.001642 | 3.29E−07 | 1.60E−06 | 2.50E−10 | 1.99E−17 | **3.81E−154** |
| 2 | best | 4.02311 | 23.3265 | 28.7236 | 0.038619 | 24.2272 | 0.113483 | 23.3448 | 0.002302 | 0.185628 | 0.285918 | **5.70E−24** | 3.50E−13 |
| | worst | 95.4854 | 26.4443 | 31.9758 | 0.067259 | 39.6549 | 0.99761 | 32.6799 | **0.004364** | 0.989079 | 1.49014 | 8.56207 | 6.13598 |
| | mean | 27.133 | 25.8494 | 30.34 | 0.055504 | 33.7613 | 0.357173 | 27.1604 | **0.003269** | 0.366328 | 0.735802 | 0.330907 | 0.453617 |
| | sd | 23.7975 | 0.607571 | 0.961172 | 0.007542 | 5.93207 | 0.309672 | 3.4977 | **0.000627** | 0.30763 | 0.243995 | 1.54065 | 1.44316 |
| 3 | best | 5.07E−06 | 2.71E−12 | 0.060012 | 0.001028 | 0.322504 | 0.000134 | 0.175958 | 0.00093 | 9.35E−05 | **0** | 5.95E−287 | **0** |
| | worst | 0.000135 | 0.001832 | 0.237043 | 0.002791 | 0.550335 | 0.850951 | 0.371275 | 0.002275 | 0.679043 | 1.44E−07 | 3.09E−06 | **1.39E−83** |
| | mean | 2.85E−05 | 0.000137 | 0.114584 | 0.001676 | 0.413367 | 0.302892 | 0.248386 | 0.001326 | 0.2167 | 9.40E−09 | 1.03E−07 | **4.64E−85** |
| | sd | 2.63E−05 | 0.000454 | 0.049979 | 0.000405 | 0.061947 | 0.268716 | 0.063439 | 0.000336 | 0.204856 | 2.96E−08 | 5.56E−07 | **2.50E−84** |
| 4 | best | 1430.16 | 711.09 | 7256.25 | 2065.45 | 4302.27 | 1065.99 | 3839.3 | 1172.04 | 1667.29 | 479.79 | 973.99 | **0.69** |
| | worst | 4009.78 | 2493.49 | 8048.99 | 3772.6 | 6232.18 | 3575.7 | 4976.6 | 2230.44 | 3808.23 | 8765.01 | 8428.67 | **2164.89** |
| | mean | 2880.63 | 1722.74 | 7718.99 | 2931.71 | 4970.56 | 2355.616 | 4306.47 | 1616.2 | 2483.364 | 3636.12 | 3448.09 | **992.24** |
| | sd | 665.536 | 346.134 | 251.117 | 396.32 | 505.547 | 693.659 | 324.354 | **281.819** | 484.67 | 2009.84 | 1529.11 | 752.587 |
| 5 | best | 2.01E−05 | 0.02125 | 0.031964 | 0.004579 | 0.985317 | 8.23E−07 | 0.162201 | 0.003088 | 4.06E−07 | **0** | **0** | **0** |
| | worst | 0.071629 | 0.201301 | 0.054336 | 0.006515 | 4.01342 | 5.35E−05 | 0.278935 | 0.005083 | 1.83E−05 | **0** | **0** | **0** |
| | mean | 0.015925 | 0.068628 | 0.044808 | 0.005549 | 1.94925 | 7.87E−06 | 0.217594 | 0.004098 | 4.55E−06 | **0** | **0** | **0** |
| | sd | 0.01754 | 0.035889 | 0.005258 | 0.000444 | 0.629016 | 1.08E−05 | 0.030519 | 0.000482 | 3.51E−06 | **0** | **0** | **0** |
| 6 | best | −1.43E−08 | −2.11E−09 | 0.009111 | 0.003985 | 0.159804 | 1.64588 | 0.137022 | 0.0037764 | 1.40487 | −1.52E−08 | **−2.14E−08** | −4.76E−09 |
| | worst | 4.00005 | 8.02E−11 | 0.012609 | 0.004652 | 0.215385 | 3.77569 | 0.177311 | 0.004444 | 3.08984 | −1.42E−14 | **−2.07E−08** | −3.46E−09 |
| | mean | 0.393583 | −3.62E−10 | 0.010757 | 0.00441 | 0.192516 | 2.74361 | 0.158296 | 0.004192 | 2.51587 | −1.77E−09 | **−2.14E−08** | −4.72E−09 |
| | sd | 0.887319 | 5.25E−10 | 0.000862 | 0.00016 | 0.014027 | 0.430395 | 0.010487 | 0.000169 | 0.394677 | 4.42E−09 | **1.75E−10** | 2.36E−10 |
| 7 | best | 0.000649 | 0.001728 | 0.018116 | 0.003075 | 0.042188 | 2.70E−06 | 0.022063 | 0.000232 | 7.90E−05 | 1.50E−06 | 7.50E−07 | **6.40E−08** |
| | worst | 0.002563 | 0.043489 | 0.052811 | 0.008607 | 0.329453 | 0.001255 | 0.122512 | 0.000797 | 0.001308 | 0.00249 | 0.001102 | **0.000455** |
| | mean | 0.001312 | 0.008538 | 0.030278 | 0.005395 | 0.109827 | 0.000615 | 0.051377 | 0.000532 | 0.000522 | 0.000634 | 0.000217 | **4.33e−05** |
| | sd | 0.000442 | 0.007738 | 0.00889 | 0.001207 | 0.065303 | 0.000357 | 0.024771 | 0.000105 | 0.000312 | 0.000741 | 0.000299 | **0.000103** |
| 8 | best | 5.30645 | 0.004985 | 7.80848 | 0.000965 | 19.1848 | 6.28E−05 | 13.4142 | 3.00E−05 | 4.60E−05 | **0** | **0** | **0** |
| | worst | 45.1519 | 0.039664 | 16.5551 | 0.001177 | 44.8064 | 0.019243 | 28.1859 | 3.77E−05 | 0.040512 | **0** | **0** | **0** |
| | mean | 22.1346 | 0.01158 | 12.1611 | 0.001082 | 31.7436 | 0.006696 | 20.667 | 3.40E−05 | 0.005412 | **0** | **0** | **0** |
| | sd | 8.62544 | 0.006953 | 2.0392 | 5.77E−05 | 7.30051 | 0.005976 | 4.8378 | 1.97E−06 | 0.008099 | **0** | **0** | **0** |
| 9 | best | 0.000371 | 0.022325 | 0.004768 | 0.000616 | 1.20685 | **−4.44E−16** | 1.21979 | 0.000601 | **−4.44E−16** | **−4.44E−16** | **−4.44E−16** | **−4.44E−16** |
| | worst | 0.002949 | 0.068854 | 0.035269 | 0.000646 | 1.6977 | **−4.44E−16** | 1.6212 | 0.000644 | **−4.44E−16** | **−4.44E−16** | **−4.44E−16** | **−4.44E−16** |
| | mean | 0.001444 | 0.036204 | 0.025515 | 0.000634 | 1.4192 | **−4.44E−16** | 1.37873 | 0.000631 | **−4.44E−16** | **−4.44E−16** | **−4.44E−16** | **−4.44E−16** |
| | sd | 0.000698 | 0.01133 | 0.007693 | 5.91E−06 | 0.127772 | **0** | 0.094204 | 7.96E−06 | **0** | **0** | **0** | **0** |

Additionally, in Table 6 the results of the comparison between COBRA-SHA and the other mentioned algorithms according to the Mann-Whitney statistical test with significance level $p = 0.01$ are presented. The same notations as in Table 3 are used in Table 6. The results of the Mann-Whitney statistical test are presented in Figures 4 and 5.

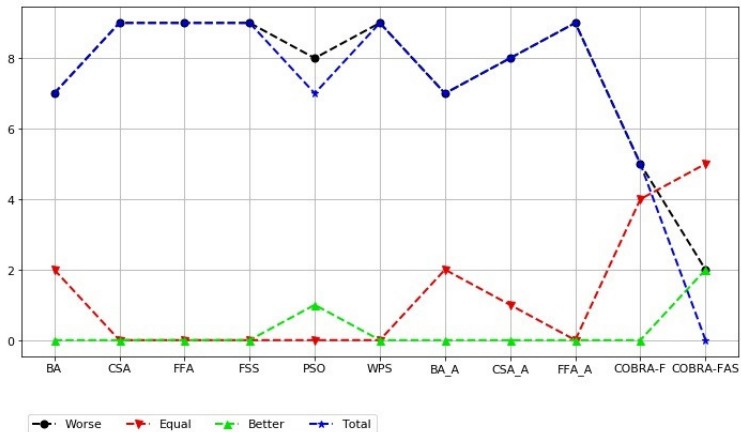

**Figure 4.** Results of the Mann-Whitney statistical test with $p = 0.01$, comparison of COBRA-SHA with other approaches (SET-2, $D = 10$).

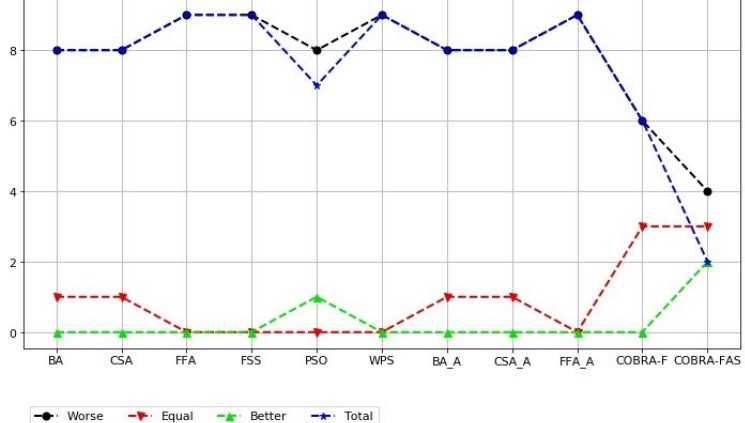

**Figure 5.** Results of the Mann-Whitney statistical test with $p = 0.01$, comparison of COBRA-SHA with other approaches (SET-2, $D = 30$).

**Table 6.** Results of the Mann-Whitney statistical test with $p = 0.01$ for SET-2, comparison of COBRA-SHA with other approaches.

| $D$ | | PSO | WPS | FSS | CSA | FFA | BA | FFA-a | CSA-a | BA-a | COBRA-f | COBRA-fas |
|---|---|---|---|---|---|---|---|---|---|---|---|---|
| | + | 8 | 9 | 9 | 9 | 9 | 7 | 9 | 8 | 7 | 5 | 2 |
| 10 | = | 0 | 0 | 0 | 0 | 0 | 2 | 0 | 1 | 2 | 4 | 5 |
| | − | 1 | 0 | 0 | 0 | 0 | 0 | 0 | 0 | 0 | 0 | 2 |
| | + | 8 | 9 | 9 | 8 | 9 | 8 | 9 | 8 | 8 | 6 | 4 |
| 30 | = | 0 | 0 | 0 | 1 | 0 | 1 | 0 | 1 | 1 | 3 | 2 |
| | − | 1 | 0 | 0 | 0 | 0 | 0 | 0 | 0 | 0 | 0 | 3 |
| Total | | 14 | 18 | 18 | 17 | 18 | 15 | 18 | 16 | 15 | 11 | 1 |

In addition, all the stated algorithms were compared with the proposed modification COBRA-SHA according to the Friedman statistical test. The obtained results are demonstrated in Figures 6 and 7. The following notations were used in Figures 6 and 7: COBRA-fas was denoted as "C-FAS" and for COBRA-SHA the notation "C-SHA" was used.

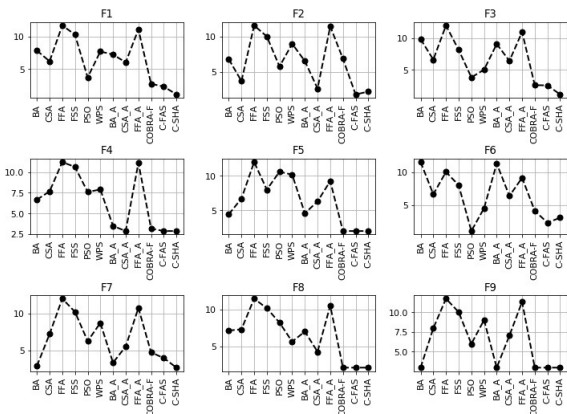

**Figure 6.** Results of the Friedman statistical test for SET-2 ($D = 10$).

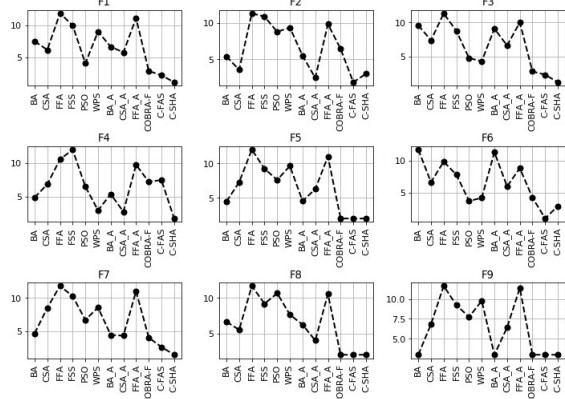

**Figure 7.** Results of the Friedman statistical test for SET-2 ($D = 30$).

It was again established that the results obtained by the proposed approach are statistically better according to the Friedman and Mann-Whitney tests than the results obtained by the mentioned biology-inspired algorithms (PSO, WPS, FSS, FFA, CSA and BA) and their modifications with the external archive (FFA-a, CSA-a, BA-a). Moreover, COBRA-SHA statically outperformed the fuzzy-controlled COBRA-f in 11 out of 18 cases. However, as for SET-1 there is almost no difference between the results obtained by COBRA-SHA and the similar modification COBRA-fas on functions from SET-2.

### 5.3.3. Numerical Results for SET-3

The next step was to test and compare the stated biology-inspired algorithms and their modifications by using benchmark functions from SET-3. The 16 functions with $D = 30$ used in SET-3 were taken from the CEC 2014 competition [51]. All these functions are minimization problems with a shifted and rotated global optimum, which is randomly distributed in $[-80, 80]$. The search range for all problems was $[-100, 100]$. The statistical results in terms of mean, standard deviation and best solution of different algorithms for functions from CEC 2014 are listed in Table 10. The best results are shown in bold.

From Table 7, it can be observed that the COBRA-SHA approach in most cases performs better than the other algorithms included in the comparison in terms of the mean value. To be more specific, this happened for the first three unimodal functions $f_1$, $f_2$ and $f_3$. Moreover, for function $f_2$ COBRA-SHA outperformed the other algorithms by all criteria. However, for $f_1$ and $f_3$ the best results (out of 51 program runs) were found by the COBRA-fas approach.

**Table 7.** Minimization results of 16 benchmark functions from SET-3 for compared algorithms.

| $f$ | | PSO | WPS | FSS | CSA | FFA | BA | FFA-a | CSA-a | BA-a | COBRA-f | COBRA-fas | COBRA-SHA |
|---|---|---|---|---|---|---|---|---|---|---|---|---|---|
| | mean | 8.39E+07 | 3.98E+06 | 1.51E+09 | 1.19E+08 | 1.21E+09 | 9.61E+07 | 4.35E+08 | 9.68E+07 | 4.13E+07 | 3.29E+06 | 2.90E+06 | **8.04E+05** |
| 1 | sd | 8.02E+07 | 4.58E+06 | 4.12E+08 | 3.48E+07 | 2.33E+08 | 9.70E+07 | 2.02E+08 | 6.10E+06 | 2.15E+07 | 1.19E+06 | 2.96E+06 | **4.52E+05** |
| | best | 3.85E+06 | 3.37E+05 | 7.10E+08 | 5.95E+07 | 7.26E+08 | 2.84E+07 | 3.02E+08 | 8.89E+07 | 1.07E+07 | 1.17E+06 | **1.60E+05** | 2.89E+05 |
| | mean | 7.21E+07 | 192442 | 1.38E+09 | 1.17E+08 | 1.10E+09 | 6.02E+07 | 4.68E+08 | 5.71E+07 | 4.03E+07 | 93605.64 | 1.97219 | **0.998929** |
| 2 | sd | 8.41E+07 | 90259.2 | 3.43E+08 | 3.26E+07 | 2.05E+08 | 6.46E+07 | 2.08E+08 | 1.29E+07 | 2.95E+07 | 1.43E+05 | 5.09984 | **2.037952** |
| | best | 2.21E+06 | 70124 | 6.55E+08 | 4.28E+07 | 6.67E+08 | 2.84E+07 | 3.14E+08 | 4.70E+07 | 2.62E+07 | 7012.77 | 0.000436 | **2.19E-05** |
| | mean | 12214.1 | 3026.72 | 8982.22 | 3591.59 | 3556.53 | 3275.33 | 3076.88 | 2458.54 | 2741.96 | 2336.28 | 2412.75 | **1085.76** |
| 3 | sd | 103.24 | 2128.7 | 7933.59 | 822.848 | 2530.09 | 2423.51 | 1832.37 | **36.1531** | 2507.81 | 2143.46 | 2682.25 | 1553.823 |
| | best | 131.23 | 141.635 | 681.498 | 1934.53 | 734.757 | 431.92 | 252.753 | 2402.45 | 200.133 | 76.757 | **0.177521** | 17.579 |
| | mean | 593.719 | 101.177 | 460.351 | 689.687 | 113.594 | 489.229 | 97.9647 | 416.944 | 388.562 | 142.139 | 83.8228 | **78.8988** |
| 4 | sd | 466.567 | 33.5444 | 404.367 | 143.184 | 28.6232 | 208.082 | **23.035** | 105.033 | 173.892 | 48.0123 | 32.0827 | 42.76286 |
| | best | 113.053 | 22.2891 | 81.996 | 355.984 | 90.3722 | 27.3223 | 82.7883 | 356.954 | 2.75597 | 20.6071 | 4.44291 | **0.086259** |
| | mean | 20.6853 | 20.23312 | 20.8368 | 21.2328 | 20.9926 | 20.9439 | 20.2327 | 20.9925 | 20.7586 | **20.0379** | 20.2732 | 20.155 |
| 5 | sd | 0.238867 | 0.315897 | **0.053797** | 0.065782 | 0.054457 | 0.135402 | 0.268122 | 0.069387 | 0.080094 | 0.112778 | 0.402076 | 0.281863 |
| | best | 20.0831 | 20.0355 | 20.6212 | 21.0601 | 20.8095 | 20.6953 | 20.0214 | 20.8928 | 20.6542 | 20.0001 | 20.0014 | **19.9997** |
| | mean | 18.8283 | 18.8698 | 45.1587 | 48.0122 | 41.2204 | 43.0776 | 40.9472 | 38.8478 | 38.5239 | 18.2647 | 15.1855 | **13.5046** |
| 6 | sd | 3.42198 | 4.00268 | 1.60656 | 1.65627 | **1.55277** | 2.36256 | 1.70132 | 2.00784 | 1.76074 | 4.1308 | 2.96969 | 1.99827 |
| | best | 10.7812 | **8.16276** | 41.887 | 44.145 | 36.8078 | 38.0094 | 35.683 | 37.5067 | 36.5252 | 10.6881 | 8.38367 | 8.86692 |
| | mean | 51.8866 | 0.518515 | 58.6923 | 0.753269 | 83.4212 | 1.12894 | 58.9804 | 0.592204 | 1.08989 | 0.411581 | 0.104303 | **0.003761** |
| 7 | sd | 25.0887 | 0.308391 | 24.1262 | 0.105676 | 42.0467 | 0.189258 | 30.6096 | 0.084048 | 0.153035 | 0.116403 | 0.58717 | **0.005962** |
| | best | 0.826791 | 0.045837 | 24.0078 | 0.505352 | 7.68658 | 0.544682 | 4.82137 | 0.524759 | 0.733626 | 0.207174 | **0** | 1.99E-06 |
| | mean | 62.388 | 96.4317 | 67.2538 | 48.4182 | 75.2201 | 49.3641 | 63.7288 | 46.6257 | 49.4966 | 8.22026 | 14.9715 | **7.70399** |
| 8 | sd | 24.4177 | 49.3056 | 17.4248 | 3.31222 | 21.842 | 3.56256 | 28.317 | 4.47713 | 2.79193 | 17.0268 | **2.49395** | 5.69825 |
| | best | 22.9252 | 39.7987 | 35.0188 | 39.1922 | 39.5088 | 39.9211 | 31.9843 | 33.9115 | 45.1326 | 1.00027 | 8.14899 | **0.025269** |
| | mean | 127.545 | 608.887 | 505.51 | 498.126 | 370.515 | 207.649 | 308.54 | 489.142 | 112.552 | 123.304 | 143.374 | **93.82118** |
| 9 | sd | 31.7914 | 44.5918 | 34.5337 | 39.9579 | 25.8412 | 39.6209 | 24.3239 | 44.33 | 28.283 | 38.3498 | 55.1968 | **23.69127** |
| | best | 71.7975 | 512.787 | 439.895 | 401.902 | 302.235 | 107.96 | 259.269 | 354.224 | 61.2855 | 55.105 | 69.082 | **48.7532** |
| | mean | 2174.14 | 2177.71 | 2092.54 | 3066.58 | 3368.45 | 1606.31 | 2251.61 | 2782.38 | 1563.79 | 789.262 | 773.36 | **440.916** |
| 10 | sd | 530.221 | 1979.64 | 704.005 | 362.634 | 948.916 | 67.3303 | 566.22 | 352.359 | **42.8626** | 829.996 | 1030.89 | 308.284 |
| | best | 1184.98 | 420.536 | 744.611 | 2060.35 | 1867.1 | 1422.06 | 1181.37 | 1810.78 | 1461.74 | 16.8956 | **0.383125** | 7.76201 |
| | mean | 3283.8 | 2661.23 | 7360.36 | 4232.33 | 4048.15 | 3430.39 | 3375.72 | 3916.73 | 3173.48 | 3063.51 | 2640.34 | **2144.361** |
| 11 | sd | 676.481 | 503.675 | 350.278 | 377.418 | 382.531 | 541.859 | 314.884 | 361.479 | 405.867 | 537.292 | 528.1465 | **329.8605** |
| | best | 2158.17 | **1393.71** | 6469.71 | 3514.68 | 3146.97 | 2454.84 | 2474.3 | 3227.44 | 2724.36 | 1843.91 | 1488.29 | 1408.18 |

**Table 7.** *Cont.*

| $f$ | | PSO | WPS | FSS | CSA | FFA | BA | FFA-a | CSA-a | BA-a | COBRA-f | COBRA-fas | COBRA-SHA |
|---|---|---|---|---|---|---|---|---|---|---|---|---|---|
| | mean | 1.38976 | 0.691141 | 1.92891 | 0.526562 | 3.64401 | 0.54154 | 2.30959 | 0.430085 | 0.417287 | 0.287138 | 0.66262 | **0.244787** |
| 12 | sd | 0.621095 | 0.138272 | 0.334828 | 0.11541 | 0.553865 | 0.133514 | 0.477087 | 0.409838 | **0.082642** | 0.112853 | 0.52344 | 0.184052 |
| | best | 0.251804 | 0.342053 | 1.21756 | 0.256158 | 2.22435 | 0.303577 | 1.36058 | **0.084931** | 0.316876 | 0.116226 | 0.106171 | 0.111626 |
| | mean | 0.948196 | 0.572278 | 0.893216 | 0.567708 | 0.791244 | 0.886569 | 0.746248 | 0.515527 | 0.90204 | 0.508758 | 0.506873 | **0.386105** |
| 13 | sd | 0.56097 | **0.039752** | 0.351253 | 0.040777 | 0.069577 | 0.075431 | 0.068078 | 0.097557 | 0.057402 | 0.112616 | 0.116107 | 0.103661 |
| | best | 0.507178 | 0.480567 | 0.613709 | 0.486725 | 0.703318 | 0.713484 | 0.598444 | 0.308969 | 0.78502 | 0.290496 | 0.277532 | **0.19549** |
| | mean | 1.10203 | 0.809976 | 0.939455 | 0.549019 | 1.81242 | 1.61733 | 1.18273 | 0.550761 | 1.62775 | 0.289925 | 0.512543 | **0.272963** |
| 14 | sd | 0.761013 | 1.34069 | 0.404122 | 0.076461 | 1.13061 | 0.190428 | 0.769868 | 0.084641 | 0.170442 | 0.093759 | 0.296611 | **0.04739** |
| | best | 0.583866 | 0.258488 | 0.516975 | 0.341437 | 0.595286 | 1.08681 | 0.500411 | 0.368413 | 1.17583 | **0.135345** | 0.197402 | 0.190261 |
| | mean | 16.4881 | 11.1549 | 13.1341 | 18.3757 | 38.1927 | 20.876 | 28.0316 | 18.1106 | 21.5514 | 14.8516 | 10.2203 | **10.0955** |
| 15 | sd | 33.3129 | 3.26455 | **0.175324** | 2.88225 | 87.3353 | 2.79681 | 62.2056 | 3.03703 | 2.39859 | 4.82808 | 3.90424 | 4.73805 |
| | best | 3.73779 | 6.22384 | 12.8651 | 11.7077 | **0.677848** | 14.2342 | 1.30462 | 10.0503 | 16.132 | 7.92992 | 4.05726 | 4.13327 |
| | mean | 11.8611 | 10.68171 | 13.2156 | 13.5836 | 13.4623 | 13.5083 | 12.9707 | 13.4556 | 13.1031 | 11.5425 | 12.3751 | **9.99059** |
| 16 | sd | 0.439775 | 0.766201 | 0.228776 | 0.202326 | 0.272579 | 0.32676 | **0.191361** | 0.208746 | 0.363715 | 0.454196 | 0.905579 | 1.13569 |
| | best | 10.8519 | 8.32414 | 12.3006 | 13.0718 | 12.8603 | 12.7806 | 12.6219 | 13.0652 | 12.745 | 10.6312 | 10.4191 | **7.30814** |

Regarding the multi-modal functions $f_4$, $f_8$, $f_9$, $f_{13}$ and $f_{16}$, COBRA-SHA was able to outperform all the biology-inspired algorithms included in the comparison in terms of mean and best values. The COBRA-SHA modification performed better than the other algorithms for the rest of the multi-modal functions (namely $f_6$, $f_7$, $f_{10}$, $f_{11}$, $f_{12}$, $f_{14}$ and $f_{15}$) except the fifth benchmark problem, but it was able to find the best value for $f_5$. The fuzzy-controlled COBRA-f found the best solution for function $f_{14}$ and gave the best mean value for function $f_5$. As with the COBRA-SHA modification, COBRA-fas gave the best values for functions $f_7$ and $f_{10}$. For functions $f_6$ and $f_{11}$, the best values were found by the WPS algorithm, while for function $f_{15}$ it was found by the FFA algorithm. Finally, the modification CSA-a was able to achieve the best value for function $f_{12}$.

The results of the comparison between COBRA-SHA and the other mentioned algorithms according to the Mann-Whitney statistical test with significance level $p = 0.01$ are presented in Table 8 (the same notations are used). The results of the Mann-Whitney statistical test are presented in Figure 8. Then all the stated algorithms were compared with the proposed modification COBRA-SHA according to the Friedman statistical test. The obtained results are demonstrated in Figure 9 (the same notations as in Figures 6 and 7 are used).

**Table 8.** Results of the Mann-Whitney statistical test with $p = 0.01$ for SET-3, comparison of COBRA-SHA with other approaches.

|       | PSO | WPS | FSS | CSA | FFA | BA | FFA-a | CSA-a | BA-a | COBRA-f | COBRA-fas |
|-------|-----|-----|-----|-----|-----|----|-------|-------|------|---------|-----------|
| $+$   | 15  | 15  | 16  | 16  | 15  | 16 | 15    | 15    | 16   | 12      | 11        |
| $=$   | 1   | 1   | 0   | 0   | 1   | 0  | 1     | 1     | 0    | 4       | 5         |
| $-$   | 0   | 0   | 0   | 0   | 0   | 0  | 0     | 0     | 0    | 0       | 0         |
| Total | 15  | 15  | 16  | 16  | 15  | 16 | 15    | 15    | 16   | 12      | 11        |

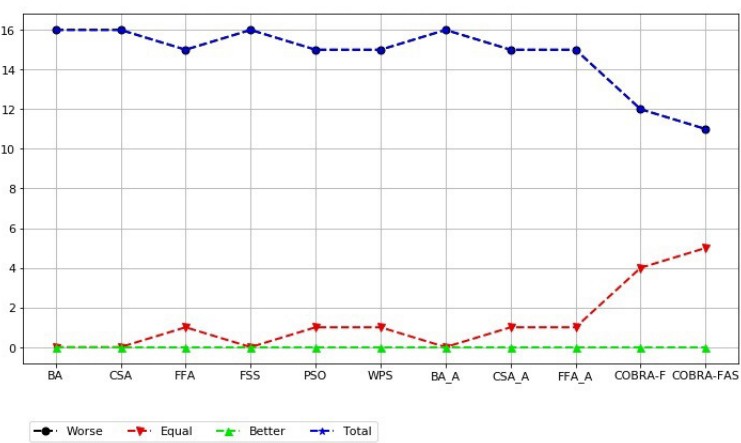

**Figure 8.** Results of the Mann-Whitney statistical test with $p = 0.01$ for SET-3, comparison of COBRA-SHA with other approaches.

Thus, it was established that the results obtained by the proposed approach are statistically better according to the Friedman and Mann-Whitney tests than the results obtained by the mentioned biology-inspired algorithms (PSO, WPS, FSS, FFA, CSA and BA) and their modifications with the external archive (FFA-a, CSA-a, BA-a). Moreover, COBRA-SHA statically outperformed the fuzzy-controlled COBRA-f. Furthermore, the experimental results for the benchmark problems from SET-3 showed that the COBRA-SHA approach is more useful for solving complex multi-modal optimization problems than the similar modification COBRA-fas. Therefore, the workability and usefulness of the proposed COBRA-SHA algorithm were demonstrated.

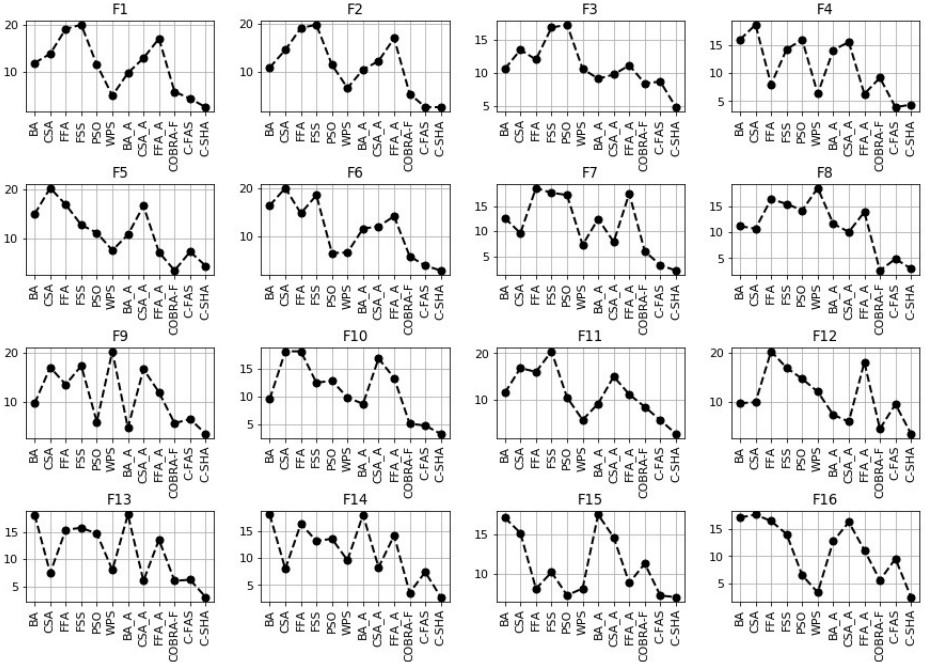

**Figure 9.** Results of the Friedman statistical test for SET-3.

### 5.3.4. Population Sizes Change

Additionally, in this study, population size changes were observed while solving benchmark problems from SET-2 and SET-3 with 10 and 30 variables. Figure 10 shows the change of the COBRA-f, COBRA-fas and COBRA-SHA component population sizes during the optimization process on three functions from SET-2 with 10 variables, namely Schwefel's function (the first column), Weierstrass's function (the second column) and Ackley's function (the third column), with the best-found fuzzy-controller parameters.

The figures on the first row demonstrate the original fuzzy-controlled COBRA-f tuning procedure behavior, the figures on the second row show the COBRA-fas modification, and finally the figures on the third row show the proposed COBRA-SHA approach. The behavior of these three tuning methods is quite different. The standard COBRA-f tends to give all resources to one component (which can be seen for Weierstrass's and Ackley's functions). However, for Schwefel's function, which is a complex optimization problem with many local minima, there was competition between the PSO and BA approaches for resources while the FFA component still had the biggest population size.

The COBRA-fas modification demonstrated similar behavior for Schwefel's function (CSA had the largest amount of resources while FSS and BA competed for "second place"). However, for Ackley's function all the components had population sizes with the number of individuals within the range [22, 23] during the optimization process. It should be noted that the same solutions were found by the COBRA-f and COBRA-fas approaches, but while COBRA-f was able to find a solution with 300 individuals throughout all populations, the COBRA-fas modification used only 133.

Finally, the proposed algorithm COBRA-SHA increased all population sizes simultaneously (but differently): each population contained at least 20 individuals. Nevertheless, the largest amounts of resources were usually given to two components: for example, in the case of Schwefel's function the winners were the FFA and BA algorithms, but for Weierstrass's function, they were the WPS and FSS algorithms. It should be noted that while solving Ackley's problem by COBRA-SHA all components had 49–51 individuals in their populations.

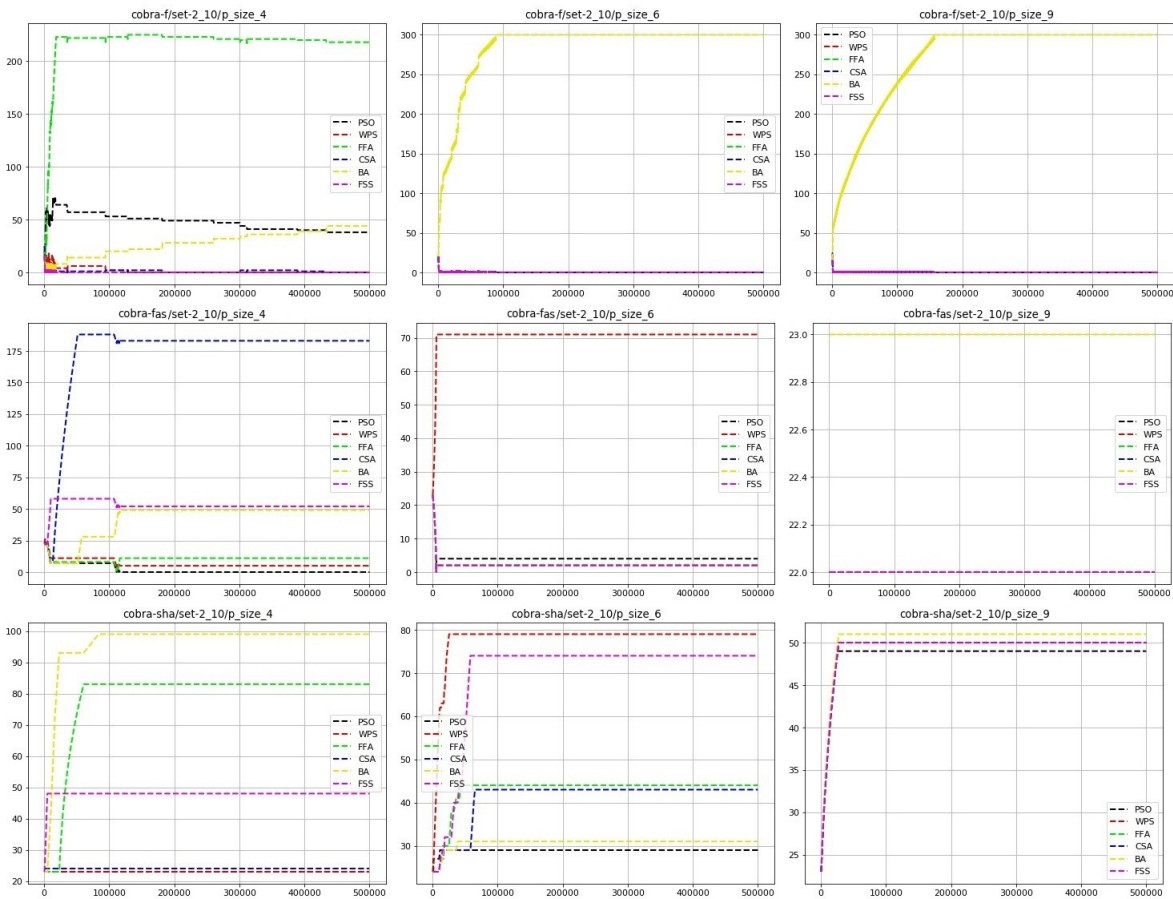

**Figure 10.** Population size changes for SET-2 with 10 variables.

Next, Figure 11 shows the change of the COBRA-f, COBRA-fas and COBRA-SHA component population sizes during the optimization process on three functions from SET-2 with 30 variables, namely the Sphere function (the first column), the Quartic function (the second column) and Ackley's function (the third column) with the best-found fuzzy-controller parameters. The algorithms demonstrate the same behavior as in the previous case (benchmark problems from SET-2 but with 10 variables).

However, let us consider the optimization process while solving the Quartic problem with the COBRA-fas approach. First, the BA algorithm appeared to be the best choice for the fuzzy controller. Thus, it increased the BA's population to 40 individuals, when other populations had minimal sizes. After that, the population sizes did not change, and only after more than 250,000 calculations was the FFA approach able to improve the optimization process, with its population size starting to increase gradually. Therefore, in the end FFA had the largest amount of resources.

Finally, Figure 12 shows the change in the COBRA-f, COBRA-fas and COBRA-SHA component population sizes during the optimization process on three functions from SET-3, namely Rotated Discus Function (the first column), Shifted and Rotated HGBat Function (the second column) and Shifted and Rotated Expanded Scaffer's F6 Function (the third column) with the best-found fuzzy-controller parameters. The first problem is unimodal, and the others are multi-modal.

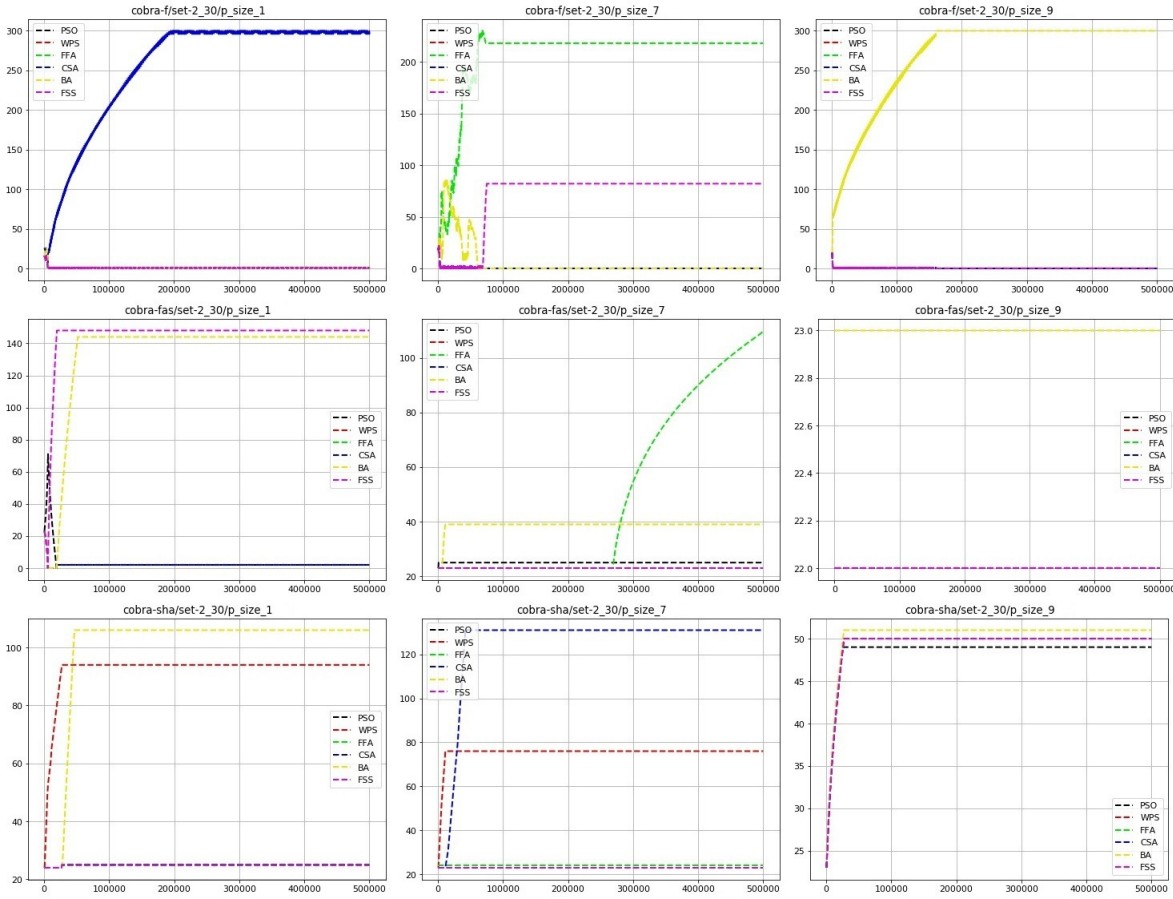

**Figure 11.** Population size changes for SET-2 with 30 variables.

The standard COBRA-f tuning method usually makes multiple oscillations, but the winning component is changed over time. The COBRA-fas method with the first problem could not make a decision regarding which component was the best during the first 150,000 calculations, but later the population of the WPS algorithm started to increase gradually and the population of the FSS algorithm became three times greater than it was initially. For the second stated problem, the PSO component appeared to be the most successful at the beginning of the optimization process. However, its population size did not change after 20,000 calculations. On the other hand, the population size of the FSS component increased after 50,000 calculations and it had the largest amount of resources by the end of the optimization process. For the last problem, as with the Discuss function, COBRA-fas could not determine the winner component algorithm at first, but then it increased the population of the FFA approach and minimized the sizes of all other populations down to zero simultaneously.

As for the COBRA-SHA modification, it did not minimize population sizes down to zero, thus, provided a more diverse set of potential solutions. Regarding the Discuss problem, even though the FFA component gave better results than other biology-inspired algorithms at first, the WPS component started to outperform it quite early. Therefore, by the end of the optimization process the WPS component had the largest amount of resources, and the FFA approach the second largest, while the populations of other components had at least 20 individuals. A similar situation can be observed for the HGBat Function with FSS and PSO components as winners. As for the last benchmark problem, during the first 70,000 calculations the population of the FFA algorithm increased and consisted of more than 100 individuals, while at the same time, the population sizes of other algorithms were close to 20 and did not change. Nevertheless, the population size of the WPS component then started to increase significantly, still delivering goal function improvements. By the end of the optimization

process, the WPS component algorithm had the largest number of individuals, yet that number was close to the number of individuals of the FFA's population.

Thus, the demonstrated cases represent different scenarios where the resource tuning is helpful, as it is able to change the algorithm structure in accordance with the current requirements.

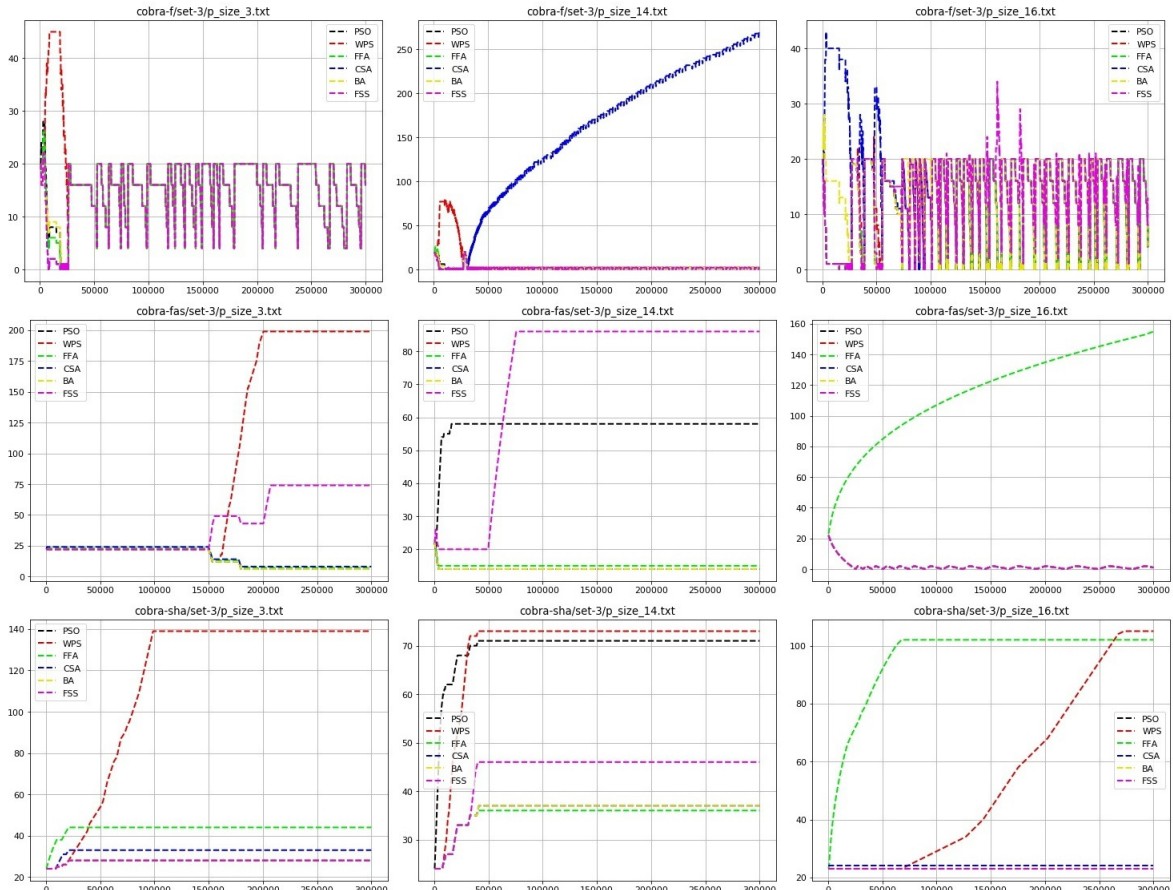

**Figure 12.** Population size changes for SET-3.

## 6. Conclusions

In this paper, a new modification of the meta-heuristic COBRA, namely the COBRA-SHA meta-heuristic, is proposed for solving real-valued optimization problems. To be more specific, a new modification is based on an alternative way of generating potential solutions. The stated technique uses a historical memory of successful positions found by individuals to guide them in different directions and thus to improve their exploration and exploitation abilities. The proposed method was applied to the components of the COBRA approach and to its basic procedures. The COBRA-SHA algorithm was tested using the three sets of benchmark functions. The experimental results show that the performance of the proposed algorithm is superior to that of the other biology-inspired algorithms in exploiting the optimum and it also has advantages in exploration.

Still, in this study several of the simplest variants of the biology-inspired component algorithms have been used for the proposed approach. Thus, further work should be focused on implementing their newer versions in the collective of the COBRA-SHA algorithm, as well as on comparisons with them. Moreover, there are still several parameters introduced for this modification, which were chosen empirically. Therefore, the performance of the COBRA-SHA approach should be tested for different parameter adaptation schemes. Moreover, the proposed modification should be applied for other optimization problem types (constrained, large-scale, multi-objective and so on).

**Author Contributions:** Conceptualization, S.A.; Methodology, S.A.; Software, S.A., V.S.; Validation, S.A., V.S., D.E.; Formal analysis, O.S.; Investigation, S.A., V.S., D.E.; Resources, S.A.; Data curation, S.A.; Writing—original draft preparation, S.A.; Writing—review and editing, V.S.; Visualization, V.S.; Supervision, O.S.; Project administration, S.A., O.S.; Funding acquisition, O.S. All authors have read and agreed to the published version of the manuscript.

**Funding:** This research was funded by the internal grant No299 for young researchers in Reshetnev Siberian State University of Science and Technology in 2020.

**Conflicts of Interest:** The authors declare no conflict of interest.

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
