# Peer review of "Success History-Based Position Adaptation in Fuzzy-Controlled Ensemble of Biology-Inspired Algorithms†"

_algorithms, doi:10.3390/a13040089_

Round 1
Reviewer 1 Report
A hybrid of several heuristic optimization methods is experimentally investigated. The co-ordination of the work of member algorithms performs a fuzzy regulator. The results of testing of the proposed hybrid algorithm with three different sets of test functions are compared with the testing results of several other hybrid algorithms. These results can be interesting for a part of the Algorithms’ audience. The paper could be recommended to be published but a revision is needed.
The hybridization of optimization algorithms has its logic which is based on properties of aimed objective functions. The theoretically based algorithms are hybridized to complement their strengths. The recognition of features of the objective function favor one or the other algorithm is the prerequisite of a reasonable hybridization. See, for example, the argumentation of hybridization of global and local search in https://doi.org/10.1016/j.cnsns.2019.104857. The properties of the objective function in your case are not considered, at least, explicitly. Thus, your criterion for complementarity of the considered algorithms should be explained properly. The selection of the member algorithm is also not sufficiently substantiated. Numerous heuristic algorithms have been proposed and named by various names from ant to wolf. However, they differ not so much algorithmically. Therefore, the selection of algorithms for hybridization should be well substantiated; otherwise it looks as ad hock success of an arbitrary choice.
It is recommended to avoid such groundless generalization as made at p.1 “Heuristic algorithms are faster and more efficient than traditional methods while solving high-dimensional complex multi-modal optimization problems, for example [3]”, although some fans of heuristic methods make such statements quite frequently. Moreover, the classification of methods into classical and heuristic is too vague. The randomization used by the heuristic methods is not a panacea; see https://doi.org/10.1007/s10898-017-0535-8.
A long list of publications is referenced in the introductions. However, the relevance of quite many of them remains not clear.
The words “This paper is an extended version of our paper published in” are repeated twice in the acknowledgement.
Author Response
Dear Reviewer,
We tried to revise our paper according to what you wrote.
Therefore, the following changes were added:
- Two mentioned studies were added to the reference list.
- We specified briefly what we meant by saying "traditional methods" and added that objective function's features are not important while using biology-inspired algorithm.
- The words “This paper is an extended version of our paper published in” were removed in the acknowledgement, so now they are not repeated there.
Reviewer 2 Report
- The authors propose a modified metaheuristic algorithm (COBRA) to improve the exploration/exploitation balance of original COBRA; the modification considers the utilization of historical memory of best positions. The proposal seems novel and exciting, but some issues must be solved in the paper to improve the overall document.
- Is it 'self-tuning' in line 107?
- The phrase in lines 184-186 must be rewritten because it seems to say that 'D' is randomly generated.
- In Algorithm 2, how is generated the individual around the algbest_i?
- In the explanation of the proposed algorithm, it would be enriching if the authors present a diagram of the whole system that includes Algorithms 1 and 2.
- Line 324 seems to have a typo.
- The statistical results presented in Table 3 are confusing; instead, it would be preferable to give the p-values. This will produce changes in lines 347-351. In the same venue, the results presented in Figure 2 are confusing; therefore, this reviewer suggests eliminating such a figure. The same logic applies for the results depicted by Figures 3-9; in other words, to the statistical analysis.
- We suggest rearranging Tables 2, 4, 5, and 7, in a more compact manner.
- The authors claim that the exploration/exploitation features of the algorithm were improved, but they do not give enough evidence in that sense (e.g., by using any diversity measure).
Author Response
Dear Reviewer,
We tried to revise our paper according to what you wrote about it. Therefore, the following changes were made:
- The word 'self-tuning' in line 107 was fixed.
- The phrase in lines 184-186 was rewritten.
- The process of the individual generating around the algbest_i in Algorithm 2 was specified.
- We think the diagram of the whole system that includes Algorithms 1 and 2 is not neccessary in this paper, also it might be too large.
- The typo in the line 324 was corrected.
- We consider the statistical results presented in Tables to be enough as the presentation of all p-values for more than 1000 statistical tests would not be informative, but might be even more confusing. Figures represent these statistical results in more compact way.
- Unfortunately that can't be done, because these tables were already reduced (median, the worst were removed). Also there are a lot of compared algorithms, and dividing one table into several might be more confusing.
- To the best of our knowledge there are no established diversity measures for meta-heuristics. Our experiments have shown that new modification of the COBRA approach allows to find better solutions as for unimodal functions (exploitation) so for complex multimodal functions (exploration).
Round 2
Reviewer 1 Report
The manuscript is revised as recommended.
Reviewer 2 Report
- After reviewing both the responses given by the authors and the document, we consider as well addressed the comments provided by this reviewer. Therefore, we suggest accepting the paper in its present form.
This manuscript is a resubmission of an earlier submission. The following is a list of the peer review reports and author responses from that submission.